EMBO
Molecular Medicine

# Glutathione peroxidase 8 negatively regulates caspase-4/11 to protect against colitis

Jye-Lin Hsu[1,2,*] [ID], Jen-Wei Chou[3], Tzu-Fan Chen[1,2], Jeh-Ting Hsu[4], Fang-Yi Su[5] [ID], Joung-Liang Lan[6], Po-Chang Wu[6,7] [ID], Chun-Mei Hu[5] [ID], Eva Y-HP Lee[8] & Wen-Hwa Lee[2,5,8,**] [ID]

## Abstract

Human caspase-4 and its mouse homolog caspase-11 are receptors for cytoplasmic lipopolysaccharide. Activation of the caspase-4/11-dependent NLRP3 inflammasome is required for innate defense and endotoxic shock, but how caspase-4/11 is modulated remains unclear. Here, we show that mice lacking the oxidative stress sensor glutathione peroxidase 8 (GPx8) are more susceptible to colitis and endotoxic shock, and exhibit reduced richness and diversity of the gut microbiome. C57BL/6 mice that underwent adoptive cell transfer of GPx8-deficient macrophages displayed a similar phenotype of enhanced colitis, indicating a critical role of GPx8 in macrophages. GPx8 binds covalently to caspase-4/11 via disulfide bonding between cysteine 79 of GPx8 and cysteine 118 of caspase-4 and thus restrains caspase-4/11 activation, while GPx8 deficiency leads to caspase-4/11-induced inflammation during colitis and septic shock. Inhibition of caspase-4/11 activation with small molecules reduces the severity of colitis in GPx8-deficient mice. Notably, colonic tissues from patients with ulcerative colitis display low levels of Gpx8 and high caspase-4 expression. In conclusion, these results suggest that GPx8 protects against colitis by negatively regulating caspase-4/11 activity.

**Keywords** caspase-4; glutathione peroxidase; inflammasome; non-canonical inflammasome; ulcerative colitis

**Subject Categories** Digestive System; Metabolism; Microbiology, Virology & Host Pathogen Interaction

## Introduction

Inflammatory caspases, including caspase-1, caspase-4, caspase-5, and caspase-11, are crucial for innate-immune defense. Caspase-1 is activated by canonical inflammasomes, consisting of intracellular nucleotide-binding oligomerization domain-like receptors (NLRs) and an adaptor protein, ASC. NLRs can be activated by cognate ligands (Lamkanfi & Dixit Vishva, 2014; Swanson *et al*, 2017), resulting in the assembly of inflammasome protein complexes and the activation of caspase-1, leading to the release of inflammatory cytokines and consequent inflammation. Thus, the dysregulation of inflammasome activity is associated with both pathogen-related and proinflammatory, non-microbial diseases, including a broad spectrum of autoinflammatory diseases (Vandanmagsar *et al*, 2011; Wen *et al*, 2011) and inflammatory bowel disease (IBD) (Caruso *et al*, 2014). In contrast to caspase-1, caspase-11 and its human counterpart, caspase-4/5, are activated independently of all known canonical inflammasome stimuli and are therefore identified as the non-canonical pathway. Caspase-11/4/5 can directly recognize bacteria LPS and undergo oligomerization, resulting in the cleavage and activation of itself (Shi *et al*, 2014) and initiating pyroptosis by cleaving the pyroptosis effector gasdermin D (GSDMD) (Kayagaki *et al*, 2015). Consequently, the activation of caspase-11 leads to the assembly of the canonical inflammasome, the NLRP3 inflammasomes, and the activation of caspase-1 (Lamkanfi & Dixit Vishva, 2014; Yang *et al*, 2015). Thus, caspase-11/4/5 are key players in sensing cytosolic lipopolysaccharide (LPS) and are responsible for LPS-induced lethality. Caspase-11 senses various bacterial infections and induces pyroptosis in infected cells, leading to the lysis of bacteria-containing vacuoles and exposure of intracellular bacteria to neutrophil-mediated killing (Miao *et al*, 2010). During endotoxemia, excessive caspase-4/11 activation causes shock. *Caspase-11*$^{-/-}$ mice are therefore highly resistant to LPS-induced septic shock (Lamkanfi & Dixit Vishva, 2014), but are particularly susceptible to *Salmonella typhimurium* (Broz *et al*, 2012) and *Burkholderia thailandensis* infection (Hagar *et al*, 2013), as well as dextran sulfate sodium (DSS)-induced colitis (Demon *et al*, 2014). Despite new insights in our understanding

1 Graduate Institute of Biomedical Sciences, China Medical University, Taichung, Taiwan
2 Drug Development Center, China Medical University, Taichung, Taiwan
3 Division of Gastroenterology and Hepatology, Department of Internal Medicine, China Medical University Hospital, Taichung, Taiwan
4 Department of Information Management, Hsing Wu University, Taipei, Taiwan
5 Genomics Research Center, Academia Sinica, Taipei, Taiwan
6 Division of Rheumatology and Immunology and Department of Internal Medicine, China Medical University Hospital, Taichung, Taiwan
7 College of Medicine, China Medical University, Taichung, Taiwan
8 Department of Biological Chemistry, University of California, Irvine, CA, USA
*Corresponding author. Tel: +88 6422052121 #7706; E-mail: jlh@mail.cmu.edu.tw
**Corresponding author. Tel: +88 6227898777; E-mail: whlee@uci.edu

of the events that trigger caspase-11 activation, how this non-canonical inflammasome pathway is modulated and whether its regulation is important in human disease remains largely unknown.

The spectrum of IBD includes inflammatory disorders of the gastrointestinal tract that already affect millions of people worldwide, and their incidence is steadily rising (Hanauer, 2006). IBD includes, ulcerative colitis (UC), a disease that is characterized by an aberrant innate-immune response of the intestine with multifactorial etiology. Associations between IBD susceptibility loci and genes involved in bacterial infection have been highlighted (Jostins et al, 2012; Cleynen et al, 2016). Since caspase-4/11 contributes critically to host defense against bacterial pathogens, dysregulation of the non-canonical pathway may lead to colitis. Importantly, reactive oxygen species (ROS) has been implicated in the regulation of caspase-11 expression (Lupfer et al, 2014) and NLRP3 inflammasome activation (Abais et al, 2015). It is likely that genes capable of modulating oxidative stress signals play a role in the non-canonical inflammasome pathway and its associated diseases.

The glutathione peroxidase (GPx) gene family plays a critical role in controlling ROS balance in organisms. Among the 8 members of the mammalian GPx family, GPx7 (i.e., NPGPx) and GPx8 share very high structural similarity and are distinguished from other GPx members by their lack of GPx enzymatic activity (Utomo et al, 2004; Chen et al, 2016). Our previous studies have shown that NPGPx acts as an oxidative stress sensor that transmits stress signals to activate downstream effectors, such as glucose-regulated protein (GRP) 78, which subsequently increases $H_2O_2$ resistance (Wei et al, 2012a). NPGPx-deficient ($NPGPx^{-/-}$) mice exhibit systemically increased ROS and show diverse phenotypes (Wei et al, 2012a; Chang et al, 2013). Conversely, observations of high levels of GPx8 expression in macrophages and the fact that $GPx8$-deficient ($GPx8^{-/-}$) mice show no apparent phenotypes under the standard housing condition indicate a possible role for GPx8 in immune defense.

In this study, we found that GPx8 in macrophages protected against DSS-induced colitis and LPS-induced sepsis and shaped the gut microbiome. Mechanistically, we uncovered that this inhibitory effect of GPx8 on caspase-4/11 activity was mediated by disulfide bonding between GPx8 and caspase-4, which occurred rapidly upon $H_2O_2$ treatment. These results indicated that GPx8 acts as a negative regulator in the non-canonical inflammasome pathway by modulating caspase-4/11 activity in response to cellular ROS levels and highlights the importance of targeting the non-canonical inflammasome in UC treatment.

## Results

### $GPx8^{-/-}$ mice exhibit exacerbated colitis and reduced gut microbiome richness and diversity

$GPx8$-deficient ($GPx8^{-/-}$) mice had the same life spans as wild-type (WT) mice housed under standard pathogen-free living conditions (Fig EV1), suggesting that $GPx8$ may only be required when the animal is under stress. $NPGPx^{-/-}$ mice appear to have a phenotypic profile that is characterized by dysfunctional immune regulation (Wei et al, 2012a; Chen et al, 2016). It is well documented that oral

DSS administration damages the epithelial monolayer lining in the large intestine and thus allows intestinal bacteria to enter the underlying tissue, provoking the infiltration of immune cells and production of proinflammatory cytokines (Chassaing et al, 2014). In view of its similarities to human UC, this model is used to study IBD pathogenesis (Perse & Cerar, 2012). Thus, we initially challenged $GPx8^{-/-}$ mice with oral DSS to investigate any potential relationship between GPx8 and immune responses. Age-, sex-, and weight-matched WT and $GPx8^{-/-}$ mice were co-housed for 3 weeks, then administered 4% DSS in their drinking water for 6 days to induce colitis (Fig 1A); mortality rates were recorded (Fig 1B–D). Whereas fewer than 50% of WT mice died during the study period, all $GPx8^{-/-}$ mice died (Fig 1B). Moreover, increased weight loss and higher clinical scores indicated by stool consistency and occult bleeding were observed in $GPx8^{-/-}$ mice compared with values in WT mice as early as day 5 and day 2, respectively (Fig 1C and D). Mean colon length was shorter and the levels of tissue damage were greater in $GPx8^{-/-}$ mice compared with WT mice treated with 4% DSS for 5 days on day 14 (Fig 1E and F).

$GPx8^{-/-}$ mice appeared to have more severe colonic inflammation than WT mice after DSS administration, as indicated by inflammatory cell infiltration (Fig 1F). Analysis of the macrophage population (F4/80$^+$ cells) by immunohistochemistry (IHC) staining revealed a significant increase in macrophages in colonic tissue of $GPx8^{-/-}$ mice (Fig 1G). Given that the inflammatory response is thought to contribute to mortality, we investigated whether this may be attributed to cytokine production. We prepared homogenates from one-third of the colon extending from the appendix and determined inflammatory cytokine production on day 14. Although interleukin (IL)-18 production by epithelial cells is critical in driving the pathologic breakdown of barrier integrity in the colitis model (Dupaul-Chicoine et al, 2010), we observed no significant difference for colonic IL-18 expression between DSS-treated WT and $GPx8^{-/-}$ mice (Fig 1H). In contrast, levels of IL-1β and IL-6 were significantly upregulated in $GPx8^{-/-}$ mice (Fig 1H). IL-6 and IL-1β are mainly produced by activated macrophages in the colon (Akira et al, 1993; Garlanda et al, 2013) and IL-1β known to be a potent inducer of IL-6 (Tosato & Jones, 1990). These results indicate that the susceptibility of DSS-induced colitis in $GPx8^{-/-}$ mice is likely to be associated with the activation of macrophages.

Overactivation of macrophages such as the NLRP3 hyper-activation is linked to alteration of the gut microbiome (Yao et al, 2017). We therefore examined the composition of gut microbiota from $GPx8^{-/-}$ mice under standard housing conditions. A beta-diversity analysis of taxonomic bacterial 16S RNA sequencing revealed significantly different fecal microbiota compositions between WT and $GPx8^{-/-}$ mice plotted either with the method of Bray–Curtis dissimilarity (Fig 1I) or with principal coordinates analysis (Fig 1J). In addition, the richness and diversity of gut bacterial species were reduced in $Gpx8^{-/-}$ mice (Fig 1K). These results imply that Gpx8 deficiency has a potential role in shaping the gut microbiome.

### GPx8-deficient macrophages contribute to the DSS-induced colitis phenotype

To test whether macrophages deficient with GPx8 play a key role in the colitis phenotype, we examined Gpx8 expression profiles in both

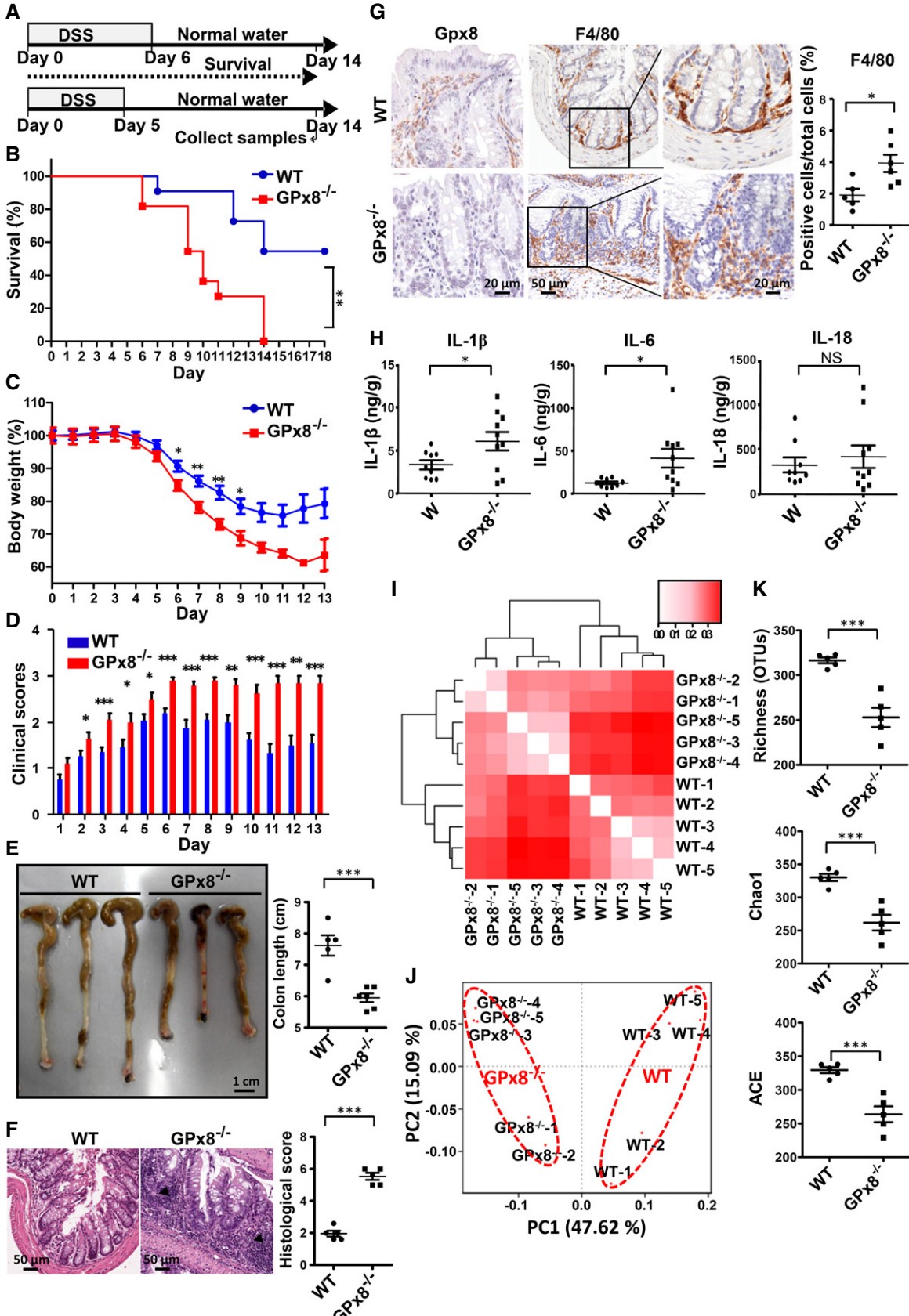

Figure 1.

**Figure 1.** $GPx8^{-/-}$ mice exhibit exacerbated colitis and reduced microbiome richness and diversity in the gut.

A   The design of DSS-induced colitis model for the survival experiment and studies for collection of colon samples. Mice were supplied with drinking water containing 4% DSS for 5 or 6 days, followed by 8–9 days of normal water. Colonic and blood samples were collected on day 14.

B–D   Mice were analyzed by Kaplan–Meier survival plot (B), the percentage of body weight (C), and the clinical scores (D) of WT and $GPx8^{-/-}$ mice with colitis induced by 4% DSS for 6 days ($n$ = 10 per group).

E–H   Colons treated with 4% DSS for 5 days were collected on day 14. For (E–G), 5–6 mice were used for each experiment. (E) Images and statistical analysis of colon length. (F) Images and semiquantitative scoring of hematoxylin and eosin staining of colon sections. The infiltrated immune cells at the transmural extensions are indicated by arrowhead. (G) Colon sections were stained with markers for macrophages (F4/80) and anti-GPx8. (H) Production of IL-1β, IL-6, and IL-18 in colon tissue lysates ($n$ = 10 per group).

I–K   Gut microbiome was shaped by Gpx8 deficiency under standard housing conditions. Bray–Curtis dissimilarity of fecal microbiota was measured by bacterial 16S RNA sequencing ($n$ = 5 per group) and is presented by a heat map (I), and principal coordinates analysis (J). In (I), the Bray–Curtis index (0–0.3) showing low to high dissimilarity is represented by the red color gradient; the deeper shades indicate higher levels of dissimilarity. (K) α-Diversity indicated bacteria species in each mouse depicted by richness (total operational taxonomic units, OTUs), the Chao1 richness estimator (Chao1), and the abundance-based coverage estimator (ACE) metrics.

Data information: In (C–K), data are presented as the mean ± SEM and calculated by using the Student's two-tailed $t$-test. *$P < 0.05$; **$P < 0.01$; ***$P < 0.001$.

murine and human colon tissues by IHC analysis. As shown in Figs 1G and EV2A, a strong positive signal was observed in the lamina propria, but not in the epithelial cells of colonic tissues. Next, we analyzed the protein expression pattern of GPx8 in different immune cell populations and found that GPx8 was predominantly expressed in macrophages, but not in T, B, or dendritic cells (Fig EV2B). Moreover, using immunofluorescence staining, GPx8 was predominantly expressed in F4/80$^+$ cells (macrophages) but not in CD103$^+$ (dendritic) cells in the gut (Fig EV2C and D), suggesting that GPx8 may have a role in macrophages.

To further determine whether GPx8-deficient macrophages contribute to the colitis phenotype, we performed an adoptive cell transfer experiment to examine the effect of GPx8-deficient macrophages on disease progression under the same gut microbiota to avoid potential interference (Fig 2A). After undergoing complete depletion of macrophages by liposomal clodronate, C57BL/6 mice received bone marrow-derived macrophages (BMDMs) from $GPx8^{-/-}$ or WT mice prior to DSS exposure (Fig 2B). Mice transplanted with $GPx8^{-/-}$ macrophages displayed greater disease severity (Fig 2C–F), and higher numbers of activated macrophages and increased cytokine production (Fig 2G and H) with DSS-induced colitis compared with mice transplanted with $GPx8^{+/+}$ macrophages. These results suggest that GPx8-deficient macrophages contribute to generating such phenotypes.

## GPx8 deficiency in macrophage potentiates caspase-11-dependent pyroptosis and exacerbates endotoxic shock

Macrophages and other innate-immune cells provide the first line of defense against microorganism invasion. To determine the signaling pathway responsible for macrophage activation and cytokine production modulated by $GPx8$, we first tested the signaling activity of all Toll-like receptors (TLRs) using BMDMs isolated from $GPx8^{-/-}$ mice. These BMDMs were treated with individual TLR agonists using a Multi-TLR Array™. The secretion of tumor necrosis factor alpha (TNF-α) and IL-6 was measured as the readout. In all TLRs examined, $GPx8^{-/-}$ BMDMs showed no detectable differences from those of WT mice (Fig EV2E), suggesting that GPx8 has little or no effect on TLR signaling pathways. We then tested whether $GPx8$ influences other pattern recognition receptors besides TLRs, such as NRP3 inflammasomes. The activation of the NLRP3 inflammasome pathways requires two signals: first, a priming signal via TLRs to increase RNA and protein expression of NLRP3, pro-IL-1β and pro-caspase-11,

which can be achieved by treating cells with TLR ligands, such as polyinosinic: polycytidylic acid [poly(I:C)](a TLR3 ligand), or LPS (a TLR4 ligand). A second signal triggering canonical NLRP3 inflammasome activation can be achieved with inflammasome inducers such as monosodium urate (MSU) crystals, nigercin, and ATP (Broz et al, 2012; Hagar et al, 2013). An alternative process for triggering NLRP3 inflammasome activation, known as the non-canonical inflammasome pathway, can be initiated by transfecting primed-BMDMs with LPS to activate caspase-11. To examine whether GPx8 is involved in the canonical or non-canonical pathways as described in Fig EV2F, we induced NLRP3 inflammasome activation under priming conditions (Fig 3A). BMDMs treated with canonical inflammasome stimuli showed no change in levels of IL-1β production, whereas $GPx8$-deficient macrophages displayed a significant increase in IL-1β production with transfected LPS (Fig 3A). Consistent with this finding, $GPx8^{-/-}$ macrophages were found to release significant amounts of IL-1β and lactate dehydrogenase (LDH), an indicator of cell death, in a dose-dependent manner with intracellular LPS stimulation (Fig 3B). Equivalently, when we substituted LPS with poly(I:C) as an alternative priming agent to exclude the potential priming signal bias, we obtained the same results (Fig 3C). Interestingly, GPx8-depleted human monocyte-derived macrophages displayed the same phenotype with increased IL-β secretion under intracellular LPS stimuli (Fig EV2G), suggesting a similarity between humans and mice in this aspect.

To further confirm the above observation, we performed a rescue experiment by ectopically expressing human GPx8 in $GPx8^{-/-}$ BMDMs for functional analysis. Overactivation of non-canonical inflammasomes in $GPx8^{-/-}$ BMDMs, including IL-1β production and LDH release, was blunted by the expression hGPx8 (Fig 3D), while the expression of hGPx8 in $GPx8^{-/-}$ BMDMs was similar to that in the WT BMDMs (Fig 3E). To further demonstrate that the non-canonical inflammasome pathway is modulated by GPx8 deficiency, the expression of full-length caspases and the amount of processed caspase-11 and caspase-1 in the supernatant were monitored by immunoblots (Fig 3F). Whereas levels of processed caspase-11 and its downstream caspase-1 were elevated in $GPx8^{-/-}$ BMDMs, the amounts of caspase-11 protein (Fig 3F, lanes 1 and 6) and mRNA (Fig 3G) in priming conditions showed no differences between WT and $GPx8^{-/-}$ BMDMs, suggesting that the enhanced activation is not caused by the increase in caspase-11 protein levels induced by priming.

Next, we tested whether GPx8 deficiency enhances the activation of the non-canonical pathway in our colitis model by analyzing the

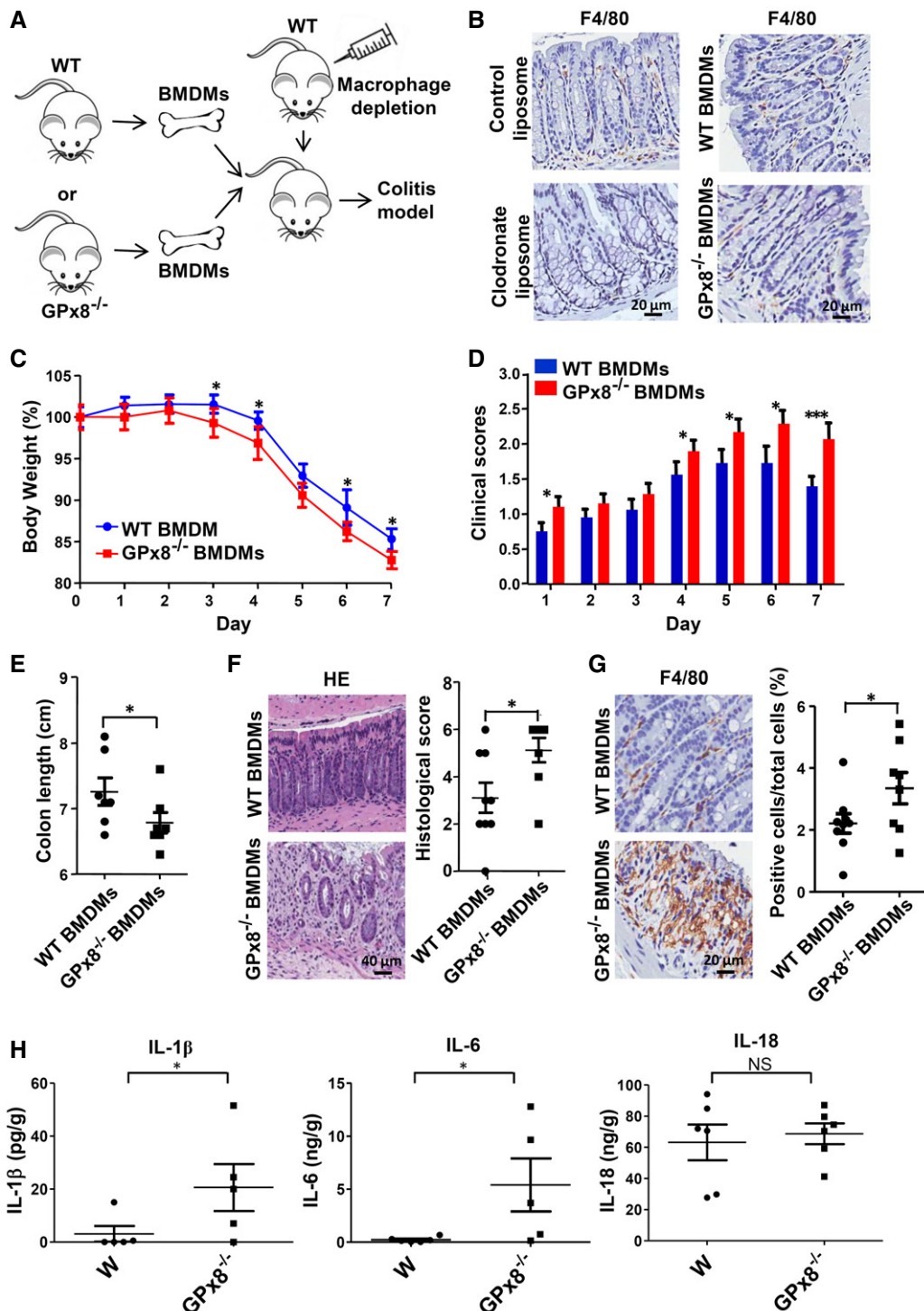

**Figure 2. GPx8-deficient macrophages enhance DSS-induced colitis.**

A    Illustration showing the experimental protocol for (B–H).
B    Colons from mice treated with clodronate or control liposomes and subjected to adoptive cell transfer after macrophage depletion was stained with markers for macrophages (F4/80).
C–H  After undergoing macrophage depletion, C57BL/6 mice received macrophages derived from WT (WT BMDMs) or *GPx8*[−/−] mice (*GPx8*[−/−] BMDMs) were under DSS-induced colitis (*n* = 6–9 per group). (C) The percentage of body weight. (D) The clinical scores of mice. (E–H) Colon samples from BMDMs transplanted mice were collected on day 7 under colitis model. (E) Statistical analysis of colon length. (F) Images and semiquantitative scoring of hematoxylin and eosin staining colon sections. (G) Colon sections stained with anti-F4/80. (H) Production of IL-1β, IL-6, and IL-18 in colon tissue lysates.

Data information: Data are presented as the mean ± SEM. *P < 0.05; ***P < 0.001 (Student's one-tailed *t*-test).

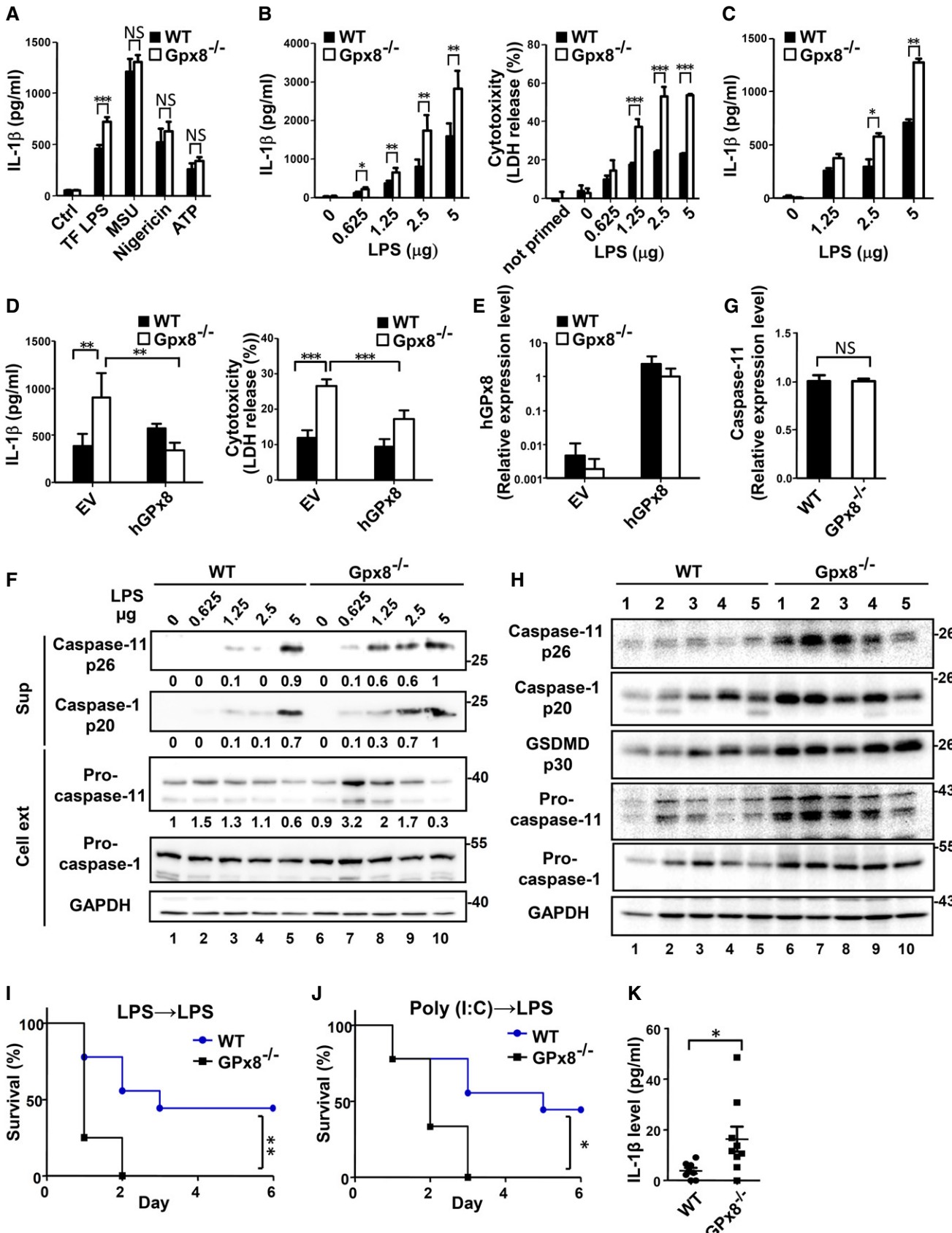

**Figure 3.**

**Figure 3. Enhancement of caspase-11 activation and increased susceptibility to endotoxic shock are associated with GPx8 deficiency.**

A   Canonical NLRP3 inflammasome activation was not altered by GPx8 deficiency. BMDMs were primed for 6 h with 0.5 µg/ml LPS and then transfected with LPS (TF LPS) for 8 h, stimulated with ATP (5 mM) for 3 h, MSU (150 µg/ml) for 8 h, or nigericin (5 µM) for 3 h. Activation of inflammasome was demonstrated by secreted IL-1β.

B   GPx8 deficiency enhanced non-canonical inflammasome activation. Activation of the non-canonical inflammasome by different amounts of cytoplasmic LPS was verified by IL-1β (left panel) and LDH release (right panel).

C   Upregulation of non-canonical inflammasome activation by GPx8 deficiency occurred independently of TLRs. BMDMs were primed for 6 h with 1 µg/ml poly(I:C) and then transfected with LPS for 8 h.

D, E   Overactivation of non-canonical inflammasome induced by GPx8 deficiency in BMDMs can be rescued by ectopically expressing human GPx8 (hGPx8). (D) Activation of non-canonical inflammasome in BMDMs with empty vector (EV) or hGPx8 was verified by IL-1β (left panel) and LDH release (right panel). (E) Expression of hGPx8 mRNA in BMDMs was confirmed by quantitative reverse transcriptase PCR.

F   GPx8 deficiency enhanced caspase-11 cleavage and activation. Levels of processed caspase-1 and caspase-11 in the supernatants (Sup) and full-length caspase-1, caspase-11, and GAPDH in the cell lysates (Cell ext) were determined by immunoblotting. Fold changes of protein expression were normalized with internal controls and are indicated below the blots.

G   RNA expression of caspase-11 was not altered by GPx8 deficiency. Caspase-11 RNA expression of BMDMs isolated from $GPx8^{-/-}$ mice was measured after LPS priming.

H   GPx8 deficiency enhanced non-canonical inflammasome activation and pyroptosis *in vivo*. Colon samples from 5 WT or $GPx8^{-/-}$ mice treated with DSS for 5 days were collected on day 7. Levels of full-length and processed caspase-1, caspase-11, and gasdermin D (GSDMD) in tissue lysates were individually analyzed by immunoblots.

I–K   $GPx8^{-/-}$ mice were more susceptible than WT mice to endotoxic shock by activation of the non-canonical inflammasome. (I) Survival of mice primed with ultra-pure LPS (*E. coli* O55:B5, 400 µg/kg) and re-challenged 6 h later with LPS (40 mg/kg) (n = 8 per group). (J) Survival of mice primed with poly(I:C) (LMW 1 mg/kg) and re-challenged 6 h later with LPS (10 mg/kg) (n = 9 per group). (K) IL-1β production levels in mice sera were measured 2 h after being primed with poly(I:C) (LMW 1 mg/kg) and re-challenged 6 h later with LPS (1 mg/kg) (n = 8–9 per group).

Data information: In (A–E and G), data are representative of at least three independent experiments and presented as the mean ± SD, n = 4–5, technical repeats. In (K), data are presented as the mean ± SEM. In (A–E, G and K), data were analyzed using the Student's two-tailed *t*-test. NS, no significance. $P > 0.05$; *$P < 0.05$; **$P < 0.01$; ***$P < 0.001$.

Source data are available online for this figure.

colons for protein expression profiles in this pathway. As shown in Fig 3H, full-length caspase-11 and caspase-1, as well as processed caspase-11 (p26), caspase-1 (p20), and GSDMD (p30, a marker of cell pyroptosis), were elevated on day 7 in colon tissue of $GPx8^{-/-}$ mice with colitis compared with samples from WT mice (Fig 3H). These results suggest that GPx8 deficiency enhances caspase-11 activation and pyroptotic cell death.

Induction caspase-11 expression *in vivo* with priming signals such as LPS or poly(I:C) (Kayagaki *et al*, 2013; Shi *et al*, 2014) renders the capase-11 pathway extremely susceptible to subsequent LPS challenges (Hagar *et al*, 2013). To validate the *in vivo* role of GPx8 (Hagar *et al*, 2013), we performed the following experiment using $GPx8^{-/-}$ and WT mice with the same genetic background, primed them with ultra-pure LPS, and re-challenged them after 6 h with a non-lethal dose of LPS. Under these conditions, 5 out of 8 WT mice survived, but all $GPx8^{-/-}$ mice died (Fig 3I), suggesting that $GPx8^{-/-}$ mice are highly susceptible to a second LPS challenge. To determine whether alternate priming pathways could substitute TLR4 *in vivo* and induce higher lethality in $GPx8^{-/-}$ mice, we primed mice with poly(I:C) and observed similar results (Fig 3J). Consistently, IL-1β production levels in mouse sera were higher in $GPx8^{-/-}$ mice primed with poly(I:C) and re-challenged with sublethal doses of LPS (Fig 3K). These results indicate that $GPx8^{-/-}$ mice are highly susceptible to endotoxic shock induced by the non-canonical inflammasome pathway through the activation of caspase-11.

## Cysteine C79 of GPx8 is required for the interaction and inhibition of caspase-4/11

GPx8 and GPx7 share a very similar structure (Utomo *et al*, 2004; Chen *et al*, 2016). It is likely that they use similar mechanisms to regulate the activity of downstream proteins by directly forming disulfide bonds (Wei *et al*, 2012a,b, 2013; Chen *et al*, 2015). To identify the covalent interacting proteins of GPx8 involved in non-

canonical inflammasome signaling, we examined whether the sensor proteins known to be activated by cytosolic LPS, caspase-11, and the human homolog, caspase-4, are the target proteins of GPx8. Indeed, results of co-immunoprecipitation assays demonstrated that Gpx8 interacted with both caspase-4 and caspase-11, but not caspase-1, a caspase that is activated by canonical inflammasome activators (Figs 4A and EV3). In addition, GPx8 and caspase-4 were co-localized in THP-1 cells (Fig 4B and Appendix Fig S1A and B). The interaction of GPx8 and caspase-4 in THP-1 cells was also demonstrated by *in situ* proximity ligation assays (PLAs). PLA-positive signals were detected in GPx8-Myc stably expressing THP-1 cells with LPS priming. In addition, these signals were fewer in these cells than in those not subjected to LPS priming, suggesting that the interaction between GPx8 and caspase-4 is induced in the presence of LPS priming signals (Fig 4C and Appendix Fig S1C–F).

To further examine whether the interaction between GPx8 and caspase-4 occurs through disulfide bonding, the immunoprecipitated complex was analyzed under reducing (+ DTT) and non-reducing (− DTT) conditions. Covalently linked high-molecular-weight complexes of GPx8 and ectopically expressed caspase-4 were detected under non-reducing conditions, but not under reducing conditions (Fig 4D), suggesting that the complex required disulfide bond formation. Since levels of oxidized GPx7 and the binding activity of it were increased in response to cellular ROS levels, we tested whether ROS modulates the interaction between GPx8 and caspase-4. A robust, GPx8-caspase-4 interaction was detected within 5 min and was maintained for another 5 min after $H_2O_2$ treatment, before gradually dissociating at 15 min (Fig 4E). These results indicated that GPx8 covalently interacts with caspase-4 by disulfide bonding rapidly in response to $H_2O_2$ treatment.

Next, we sought to determine which cysteine residue of GPx8 is involved in the bonding with caspase-4, by generating three mutants with serine substitutions: C79S and C108S, as well as C2S2, containing both C79 and C108. Co-immunoprecipitation

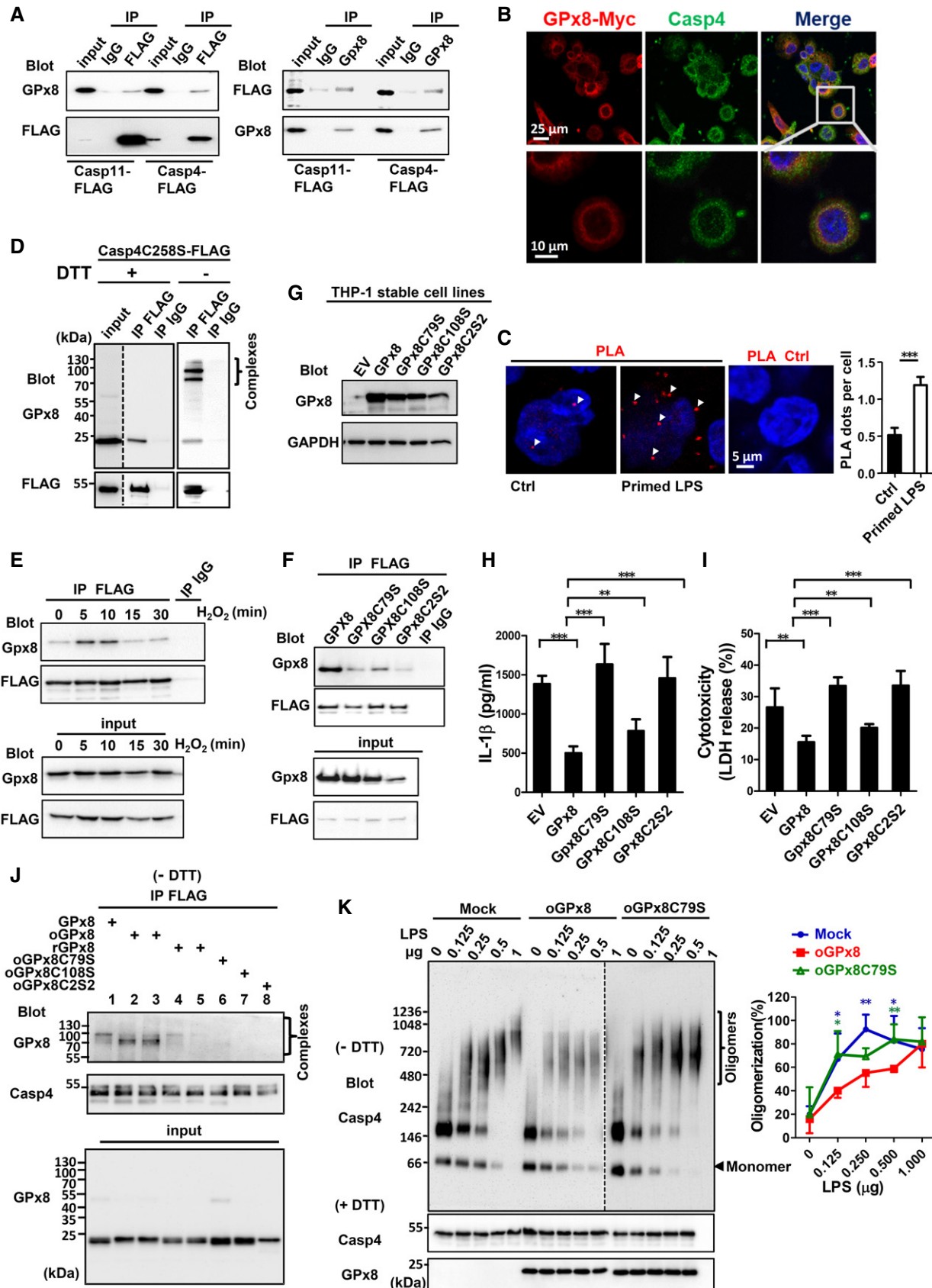

Figure 4.

◄

**Figure 4. Cysteine C79 of GPx8 is required for the interaction and inhibition of caspase-11/4.**

A   Interaction of GPx8 with human caspase-4 (Casp4) or mouse caspase-11 (Casp11) was confirmed by co-IP assays. 293T cells were co-transfected with vectors expressing FLAG-tagged Casp4 or Casp11, in addition to GPx8 expressing vectors. Cell lysate containing Casp4- or Casp11-FLAG proteins was immunoprecipitated by anti-FLAG Ab and subsequently analyzed by Western blots using anti-GPx8 and anti-FLAG Ab (left panel). GPx8 and its interacting proteins were precipitated by anti-GPx8 Ab and analyzed by the indicated Abs (right panel).

B   Co-localization of Myc-tagged GPx8 (red) and endogenous caspase-4 (green) in THP-1 cells. For details, see Appendix Fig S1.

C   Interaction between GPx8 and caspase-4 in THP-1 cells in the presence or absence of priming LPS was demonstrated by the *in situ* proximity ligation assay (PLA) (red dots, labeled by arrowheads). GPx8-Myc stably expressing THP-1 was used to detect the interaction between GPx8-Myc and endogenous caspase-4. Untreated cells (Ctrl) or cells primed with LPS (primed LPS) were incubated with anti-Myc and anti-caspase-4 Abs, according to the manufacturer's instructions, and the nucleus was stained with DAPI (blue). Results were quantified by counting at least 5 different fields with an average of 300 cells. Reactions without primary Abs were used as the control for the PLA assay (PLA Ctrl).

D–F   293T cells were co-transfected with vectors expressing FLAG-tagged caspase-4 (WT or C258S) in addition to GPx8 expressing vectors. Cell lysate was immunoprecipitated by anti-FLAG Ab and subsequently analyzed by Western blots using anti-GPx8 or anti-FLAG Ab. (D) Complexes of GPx8 and caspase-4 were demonstrated by co-IP assays. Cysteine-dependent covalent interactions of GPx8 and caspase-4 (C258S) were analyzed under non-reducing and reducing conditions. Disulfide-linked complexes of Gpx8 and caspase-4 are indicated. (E) Interactions of GPx8 and caspase-4 in response to $H_2O_2$ treatment were demonstrated by co-IP assays of lysates prepared from 293T cells treated with $H_2O_2$ (200 μM). (F) C79S, C108S, and C2S2 mutants of GPx8 abolished most of the binding to caspase-4 by co-IP assays.

G–I   C79 of GPx8 was essential for its inhibitory effect on non-canonical inflammasome activation. THP-1 cells stably expressing empty vector (EV), WT, or mutant GPx8 proteins were confirmed by immunoblotting (G). (H and I) Cells were treated with 12-O-tetradecanoylphorbol 13-acetate (PMA) (0.1 μM) and differentiated for 3 days then primed and transfected with LPS. Activation of the non-canonical inflammasome by different amounts of cytoplasmic LPS was verified by IL-1β (H) and LDH release (I).

J   *E. coli* purified oxidized GPx8 interacts with mammalian cell purified FLAG-tagged caspase-4 (C258S) to form a disulfide-linked complex (oxidized GPx8: oGPx8; reduced GPx8: rGPx8). Lanes 2 and 3, GPx8 pretreated with 1 and 5 mM $H_2O_2$, respectively. Lanes 4 and 5, Gpx8 pretreated with 50 and 200 mM DTT, respectively. Disulfide-linked complexes of recombinant Gpx8 and caspase-4 (C258S) are indicated.

K   GPx8 inhibited the activation of caspase-4 by decreasing the oligomerization of caspase-4. Oxidized GPx8 and caspase-4 (C258S) were co-incubated for 30 min. Different amounts of LPS were added to the complex and incubated for another 30 min. Oligomerized caspase-4 was analyzed by native blue PAGE according to the manufacturer's instructions. Mock: without GPx8; oxidized GPx8: oGPx8; oxidized GPx8C79S: oC79S.

Data information: All data are representative of at least three independent experiments. In (C), data are presented as the mean ± SD, *n* = 5, technical repeats. In (H and I), data are presented as the mean ± SD, *n* = 4, technical repeats. In (K), quantification of oligomers was calculated from three independent experiments and is presented as the mean ± SD of percent of oligomers/total proteins. *P < 0.05; **P < 0.01; *** P < 0.001 (Student's two-tailed *t*-test).

Source data are available online for this figure.

experiments with FLAG-tagged caspase-4 revealed that cysteine residues C79 and, to a lesser extent, C108, were required for the interaction with caspase-4 (Fig 4F). We then tested whether disulfide interactions are important for non-canonical inflammasome signaling. Human THP-1 monocytes expressing compatible levels of mutant GPx8 proteins with WT were constructed for the assay (Fig 4G). Expression of WT GPx8 in THP-1 cells inhibited intracellular LPS-induced IL-1β production and cell death (Fig 4H and I). In contrast, expression of GPx8 C79S and C2S2 mutants completely abolished all inhibitory activity (Fig 4H and I). These results confirm that GPx8 is important for inhibiting caspase-4/ 11-dependent response to cytoplasmic LPS and that the cysteine residues C79 and, to a lesser extent, C108, play an important role in this process.

To further investigate the interaction between caspase-4 and GPx8, we used purified recombinant GPx8 and caspase-4 (C258S) protease-deficient mutant protein (Shi *et al*, 2014) for *in vitro* complex formation assays. As shown in Fig 4J, a high-molecular-weight complex was identified in the presence of oxidized GPx8, but not with reduced GPx8 (Fig 4J, upper panel, lanes 2, 3, 4, and 5). Conversely, the expression of GPx8 C79S, C108S, and C2S2 mutants completely eliminated the complex formation (Fig 4J, upper panel, lanes 6, 7 and 8). These results confirm that both C79 and C108 are required for the direct interaction between GPx8 and caspase-4.

Since LPS-induced oligomerization of caspase-4 is a prerequisite for its catalytic activation, we tested whether GPx8 modulates the oligomerization activity of caspase-4. In the presence of oxidized GPx8, but not the GPx8 C79S mutant or the control, LPS-induced oligomerization of caspase-4 was reduced (Fig 4K, left panel) and the percentage of oligomers/total protein was lower (Fig 4K, right panel), implying that GPx8 inhibits the activation of caspase-4 by decreasing the oligomerization of caspase-4. Because caspase-4 can directly bind to LPS, we tested whether LPS binding activity was regulated by GPx8 (Appendix Fig S2). Unlike the oligomerization activity, oxidized GPx8 did not interfere caspase-4 LPS binding activity. These data suggest that GPx8 interacts directly with caspase-4 via disulfide bonding, which negatively regulates downstream inflammasome activation.

**GPx8 inhibits caspase-4/11 through covalent binding to cysteine 118**

GPx8 deficiency constrained non-canonical but not canonical NLRP3 inflammasome activation. We therefore aligned human and mouse caspase-1 and -4 for evidence of unique cysteine residues present only in caspase-4 and caspase-11. Two unique cysteine residues, C118 and C329, were conserved in caspase-4 and caspase-11, but not in human and mouse caspase-1 (Appendix Fig S3A), suggesting that they may be involved in disulfide bonding.

To test this possibility, we generated caspase-4 mutants with serine substitutions at the indicated cysteine sites. To verify whether caspase-4 mutants covalently interact with Gpx8, co-immunoprecipitation of caspase-4 and Gpx8 was performed and analyzed under reducing (+ DTT) and non-reducing (− DTT) conditions. The C118S mutant, not the C329S mutant, lost the greatest amount of GPx8 and caspase-4 complexes (Fig 5A). All remaining cysteine residues in caspase-4 were also tested, but none had any effects on

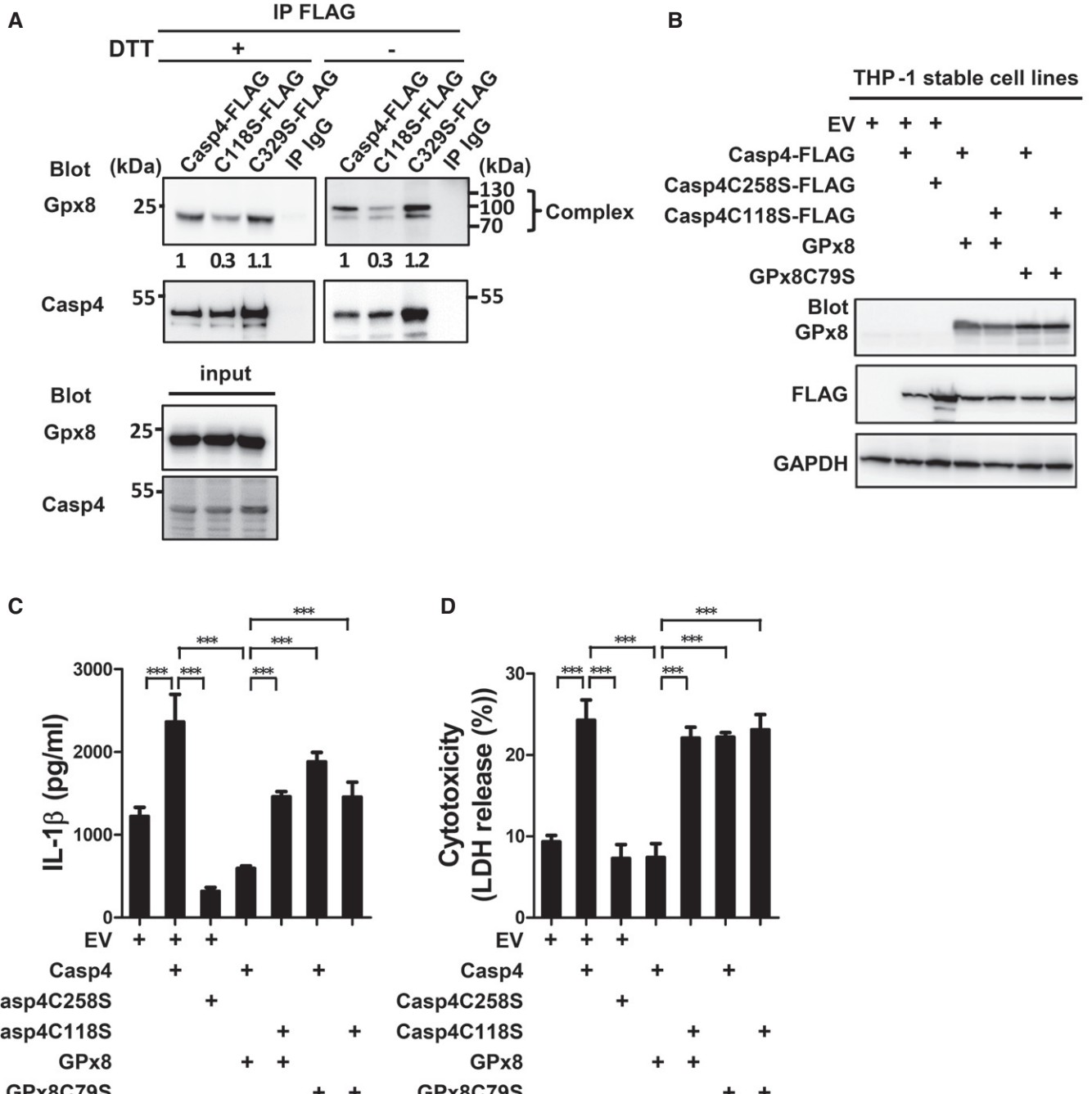

**Figure 5. GPx8 inhibits the activity of caspase-4 by covalently binding to cysteine 118.**

A  Casp4C118S mutant partially abolished the covalent disulfide-link with GPx8. Interacting GPx8 were co-immunoprecipitated with FLAG-tagged caspase-4 (WT, C118S, C329S mutants) and analyzed under reducing (+DTT) or non-reducing (−DTT) conditions. Disulfide-linked complexes of GPx8 and caspase-4 are indicated. Fold change of complexes was normalized with internal controls and indicated below blots.

B–D  C118 of caspase-4 was required for the GPx8-mediated inhibitory effect. THP-1 cells stably expressing empty vector (EV), FLAG-tagged caspase-4 (WT, C258S, C118S mutants), or GPx8 (WT and C79S mutants) were confirmed by immunoblotting (B). (C, D) Casp4C118S completely abolished the inhibition of non-canonical inflammasome signaling in response to intracellular LPS. THP-1 cells were differentiated, primed, and transfected with LPS. Activation of the non-canonical inflammasome by different quantities of cytoplasmic LPS is confirmed by the production of IL-1β (C) and LDH (D).

Data information: All data shown are representative of at least three independent experiments. In (C and D), data are presented as the mean ± SD, *n* = 4, technical repeats. ***$P < 0.001$ (Student's two-tailed *t*-test).
Source data are available online for this figure.

the disulfide bonding between GPx8 and caspase-4 (Appendix Fig S3B). We then tested whether the activity of the C118S mutant is modulated by GPx8. Human THP-1 monocytes stably expressing caspase-4 and GPx8 constructs were confirmed by immunoblotting (Fig 5B). Unlike THP-1 cells expressing WT caspase-4, which showed reduced activity in the presence of GPx8, mutant expressing caspase-4 C118S showed no difference in IL-1β production or cell death in the presence or absence of a functional GPx8 (Fig 5C and D). These results indicate that C118 of caspase-4 is important for the interaction between GPx8 and caspase-4 and that caspase-4 C118S contributes to GPx8-mediated inhibition of non-canonical inflammasome activation.

### Treatment with a caspase-4/11 inhibitor, VX-765, or a strong antioxidant, N-acetylcysteine, suppresses caspase-4/11-dependent inflammasome activation and reduces DSS-induced colitis in $GPx8^{-/-}$ mice

Excessive inflammasome activation is linked to morbidity and mortality in DSS-induced colitis (Bauer *et al*, 2010). We therefore treated $GPx8^{-/-}$ mice with VX-765, a caspase-1/4 inhibitor (Stack *et al*, 2005; Wannamaker *et al*, 2007), or N-acetylcysteine (NAC), an ROS scavenger, to confirm the importance of the ROS-modulating gene in the inflammasome pathway and colitis. Initial testing revealed that VX-765 suppressed cytosolic LPS-induced caspase-4/11 activation in $GPx8^{-/-}$ BMDMs (Fig 6A). Consistently, VX-765 treatment was also associated with lower mortality and less disease severity in mice with DSS-induced colitis (Fig 6B–D), specifically with significantly less body weight loss and lower clinical scores compared with mice that did not receive VX-765 (Fig 6C and D). Activation of caspase-11 (p26), caspase-1 (p20), GSDMD (p30), and production of IL-1β and IL-6 in colonic tissues was considerably reduced by VX-765 treatment (Fig 6E and F). Similarly, NAC treatment suppressed caspase-4/11-dependent inflammasome activation in $GPx8^{-/-}$ BMDMs (Fig EV4A) and reduced colitis severity in $Gpx8^{-/-}$ mice (Fig EV4B–E). These results indicate that the high susceptibility of $GPx8^{-/-}$ mice to DSS-induced colitis could be attributed to the activation of the non-canonical inflammasome, which exacerbates IL-1β and IL-6 production, resulting in severe colon inflammation and tissue damage.

### Colon tissues from UC patients express lower levels of GPx8 and higher levels of caspase-4

To assess the role of GPx8 and caspase-4 in clinical inflammatory disease, we examined expression levels of GPx8 and caspase-4 in colonic tissue samples from UC patients and healthy controls. We used endoscopy to visualize and obtain non-inflamed intestinal biopsies from UC patients and normal intestinal biopsies from non-IBD healthy individuals. To confirm whether GPx8 is predominantly expressed in macrophages of UC patients, colonic biopsy sections of UC patients were stained with a marker for pan-macrophage, CD68, or a monocyte/macrophage activation marker, CD163, and examined for GPx8 expression. Consistent with murine expression profiles, GPx8-expressing cells were found to be highly associated with CD68 and CD163 (Fig 7A), indicating that GPx8 is mainly expressed in colonic macrophages. Furthermore, compared with the colons from normal controls, GPx8 expression was significantly reduced in

non-inflamed colon biopsies from UC patients (Fig 7B and C). In addition, a significant difference with pairing GPx8 and caspase-4 expression in each individual was observed between patients and normal controls (Fig 7D) by Fisher's exact test. Notably, GPx7 expression did not differ between patients with UC and controls (Fig 7B and C), whereas, caspase-4 expression was significantly higher in the UC cohort, in agreement with a previous report (Flood *et al*, 2015). These results demonstrate that UC individuals have lower Gpx8 and higher caspase-4 expression in the colon, suggesting that GPx8 acts as a negative regulator by down-regulating caspase-4 activation and protecting against colitis.

## Discussion

In this communication, we found that $GPx8^{-/-}$ mice exhibited exacerbated colitis induced by DSS and were more susceptible to endotoxic shock when challenged by LPS. $GPx8$-deficient macrophages displayed enhanced caspase-11 activity, resulting in increased levels of pyroptosis and IL-1β production. GPx8 dampened caspase-4/11 activation directly through disulfide bonding mediated by cysteine 79 of GPx8 and cysteine 118 of caspase-4. Treatment with NAC, a potent antioxidant, or with VX-765, a caspase-4 inhibitor, consistently suppressed caspase 4/11-dependent inflammasome activation and reduced colitis severity in $Gpx8$-deficient mice. Significantly, a positive correlation was found between lower Gpx8 and higher caspase-4 expression in colonic tissue from UC patients. Thus, GPx8 appears to play a negative regulating role in the non-canonical inflammasome pathway in response to cellular ROS levels, implying that GPx8 impacts on the pathogenesis of IBD and relevant disease pathways, as illustrated in Fig EV5.

Many genes that are involved in the dysregulation of the inflammasome pathway are linked with autoinflammatory diseases, including IBD (Guo *et al*, 2015). Functional genetic studies have revealed a group of IBD susceptibility genes, including *NLRP3* (Villani *et al*, 2009), *ASC* (Zaki *et al*, 2010), and *caspase-1* (Zaki *et al*, 2010; Demon *et al*, 2014). Several IBD susceptibility loci identified by genomewide association studies are antioxidant genes, such as *GPx1* (Jostins *et al*, 2012) and *GPx4* (Jostins *et al*, 2012). Oxidative stress specifically induces gastroduodenal ulcers, IBD, and even gastric and colorectal cancers (Grisham, 1994; Hisamatsu *et al*, 2013). Low levels of superoxide dismutase and glutathione peroxidase in patients with active IBD (Grisham, 1994) may also reflect the importance of ROS in this disease. The link between GPx8 and IBD described in this paper is consistent with the notion that modulating the non-canonical inflammasome pathway in response to ROS contributes to the disease etiology.

It has been proposed that oxidative stress is involved in inflammasome activation (Zhou *et al*, 2010; Abais *et al*, 2015). Notably, NLRP3 contains a highly conserved disulfide bond that connects a pyrin domain with the nucleotide-binding site domain; this structure is highly sensitive to altered redox states (Abais *et al*, 2015). In addition, inflammasome activators, such as uric acid crystals, induce the dissociation of thioredoxin-interacting proteins from thioredoxin in a ROS-sensitive manner, allowing thioredoxin to bind to and activate NLRP3 (Zhou *et al*, 2010). Our discovery demonstrates that how ROS modulates the key component of the non-canonical inflammasome pathway. That is GPx8 serves as a redox

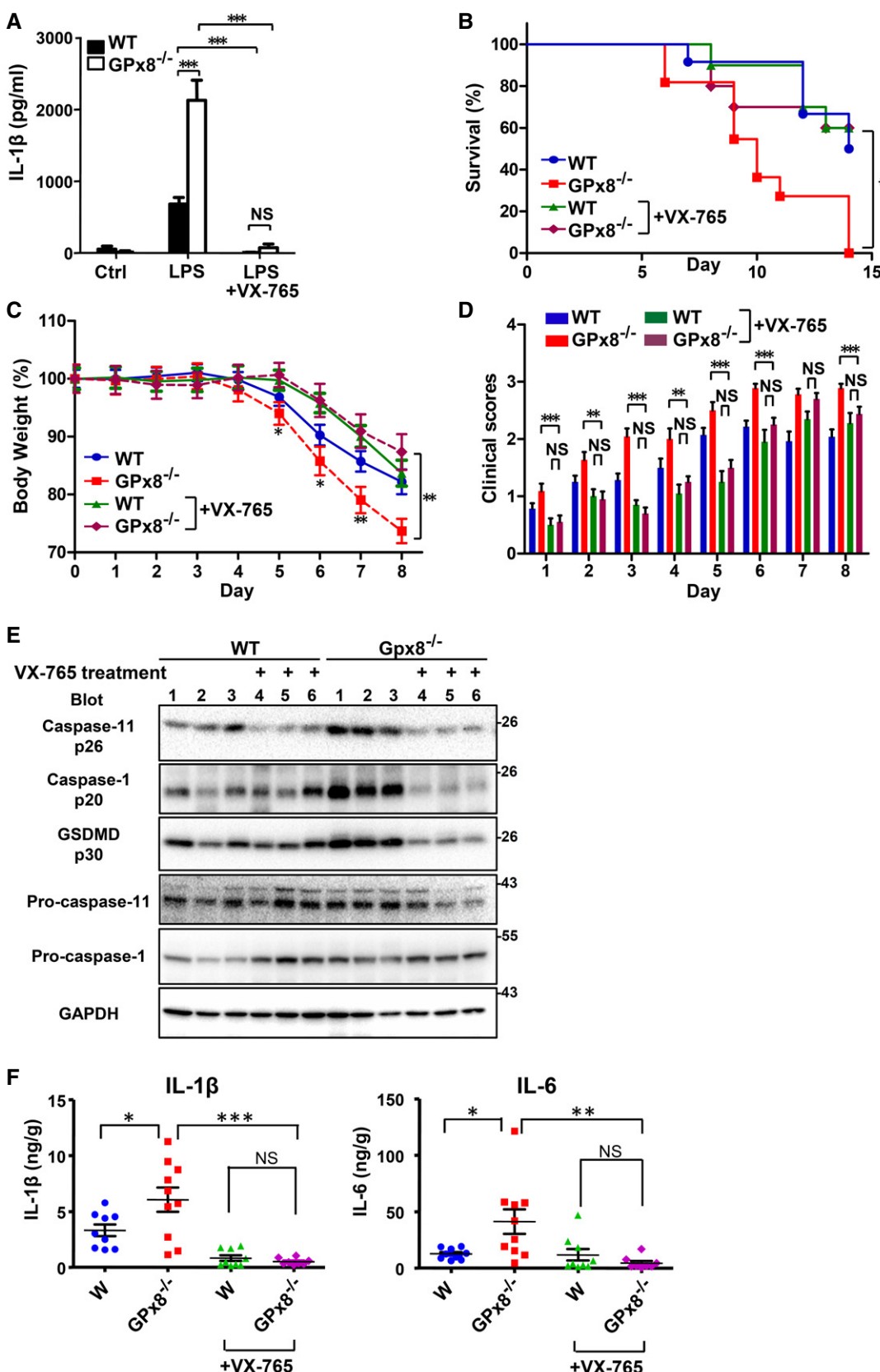

Figure 6.

**Figure 6.  The caspase-4/11 inhibitor, VX-765, suppresses DSS-induced colitis in *GPx8*⁻/⁻ mice.**

A    The caspase-4/11 and caspase-1 inhibitor, VX-765, inhibited non-canonical inflammasome activation. BMDMs isolated from mice were primed with LPS for 5–6 h, treated with 20 μM VX-765, and transfected with LPS for 12 h.

B–F  Mice were injected intraperitoneally with VX-765 (20 mg/kg) under colitis model as shown in Fig 1A and analyzed by a Kaplan–Meier survival plot (B), the percentage of body weight (C), and clinical scores (D) of mice with colitis induced by 4% DSS for 6 days (n = 10–11 per group). (E, F) Colon samples from mice treated with 4% DSS for 5 days (n = 6–10 per group). Full-length and cleavage forms of caspase-11, caspase-1, and GSDMD (E) in colon tissue lysates were analyzed by immunoblots on day 7 after DSS treatment. Production of IL-1β and IL-6 (F) in colon tissue lysates was assessed on day 14 after treatment.

Data information: In (A), data are presented as the mean ± SD, n = 5, technical repeats. In (B–F), data are presented as the mean ± SEM. In (A, C, D and F), data were analyzed using the Student's two-tailed t-test. NS: not significant (P > 0.05). *P < 0.05; **P < 0.01; ***P < 0.001.

Source data are available online for this figure.

sensor, is oxidized by ROS, and transfers inhibitory signaling to the non-canonical inflammasome initiator, caspase-4/11.

GPx is a critical peroxide-scavenging enzyme that reduces ROS levels in cells. Notably, GPx7 and GPx8 lack enzymatic activity and share highly similar amino acid sequences and domain structures. We have previously reported that GPx7 is an oxidative stress sensor that is essential for maintaining redox balance (Wei *et al*, 2012a). Unlike GPx7 deficiency, characterized by systemic defects and an overall increase in ROS levels, GPx8-deficient mice showed no differential phenotypes or life spans, implying a systemic ROS sub-relevant function. A comparison of WT and *GPx8*-deficient macrophages revealed very few differences in overall intracellular ROS levels (Appendix Fig S4), mRNA expression (Fig 3G), and protein levels (Fig 3F, lanes 1 and 6) of caspase-11. Under activation, caspase-4 and ROS start to accumulate in macrophages, GPx8 is activated (oxidized) by ROS to dampen caspase-4 oligomerization and tightly regulate caspase-4 activation. When ROS exceeds a certain level, the GPx8-caspase-4 interaction gradually decreases and caspase-4 is free to activate downstream signals (Fig 4E). Therefore, *GPx8*-macrophages exhibit increases in caspase-4 oligomerization and inflammasome activation.

Caspase-4 exists as a precursor that consists of the caspase-recruitment domain (CARD), p19 and p10 domains. Recent studies have shown that caspase-4/11 exists in the cell as a monomer (~48 kDa) and recognizes LPS via its CARD domain, which subsequently undergoes oligomerization (~600 kDa; Shi *et al*, 2014). Caspase-4 is then processed and forms an enzymatic active form of caspase with heterodimers that contain two p19 and two p10 domains. There are eight cysteine residues in caspase-4, none of which are located in the CARD domain. Since GPx8 regulates caspase-4 activity via disulfide bonding, it is more likely that GPx8 has no effect on LPS binding activity. Indeed, the LPS pull-down assay showed that GPx8 does not interfere with the LPS binding

activity of caspase-4 (Appendix Fig S2). It is more likely that GPx8 facilitates the disulfide bond formation of caspase-4 to reduce its oligomerization activity after LPS binding. Consistent with this scenario, we demonstrated that WT, and not the GPx8C79S mutant protein, is able to interact with caspase-4 and modulate its oligomerization in response to LPS. When we aligned the amino acid sequences of caspase-4, we found that the unique cysteine residue (C118) targeted by GPx8 is conserved in caspase-4 and caspase-11, but not in caspase-1, which explains how GPx8 deficiency enhances the activation of caspase-4/11 and not caspase-1. Because C118 is also highly conserved in mammals, the regulatory mechanism of GPx8 on caspase-4 may be a shared feature among different species, implying that the GPx8 interacting domain in the p20 region in caspase-4 may be important for oligomerization. Since oligomerization is a prerequisite for the catalytic activity of caspase-4, reducing the oligomerization by an inhibitory protein, such as GPx8 or small molecules, could be an alternative approach for drug development. Indeed, inhibitors have been identified that bind to a single cysteine in the dimerization interface of caspases and thereby prevent the dimerization and activation of the caspase-3 proenzyme (Hardy *et al*, 2004). In our preliminary result, we found that multiple cysteines are important for protease activity. Thus, it will be of interest to examine how caspase-4 changes structure before and after binding to LPS, and how GPx8 influences the disulfide bond formation of caspase-4 in the process.

It is acknowledged that caspase-11 is critical for protecting mice from a variety of bacterial infections (Broz *et al*, 2012; Hagar *et al*, 2013) and colitis (Demon *et al*, 2014). Since the activation of caspase-4/11 eventually leads to IL-1β production, which plays a pivotal role in regulating innate-immune responses and the homeostasis of colon epithelial cells, GPx8 deficiency may cause increased inflammation and high susceptibility to colitis. In addition, a clear correlation between levels of caspase-4 expression and

**Figure 7.  Lower GPx8 and higher caspase-4 expression in colon tissue of ulcerative colitis patients.**

A    GPx8 was predominantly expressed in macrophages of UC patients. Colon biopsies of non-inflamed regions from UC patients were stained with the pan-macrophage marker, CD68, or the monocyte/macrophage activating marker, CD163, combined with anti-GPx8. Quantification of double-positive cells was calculated from 3 individual patients and is presented as the mean ± SEM of the percentage of double-positive cells. The total cell numbers are the sum of double- and single-positive cells.

B    Relative expression levels of GPx8, GPx7, and caspase-4 in non-inflamed biopsy specimens from healthy controls (Ctrl) (n = 14) and UC patients (n = 25) are demonstrated by immunoblots.

C    Data are normalized to the mean of healthy controls and presented as the mean ± SEM. Mann–Whitney test for GPx8 and Student's two-tailed t-test for GPx7 and caspase-4 expression analysis.

D    Paired GPx8 and caspase-4 expression from each individual are linked by lines and presented as the mean ± SEM. The P-value was analyzed by Fisher's exact test.

Data information: NS: not significant (P > 0.05). *P < 0.05; **P < 0.01; ***P < 0.001.

Source data are available online for this figure.

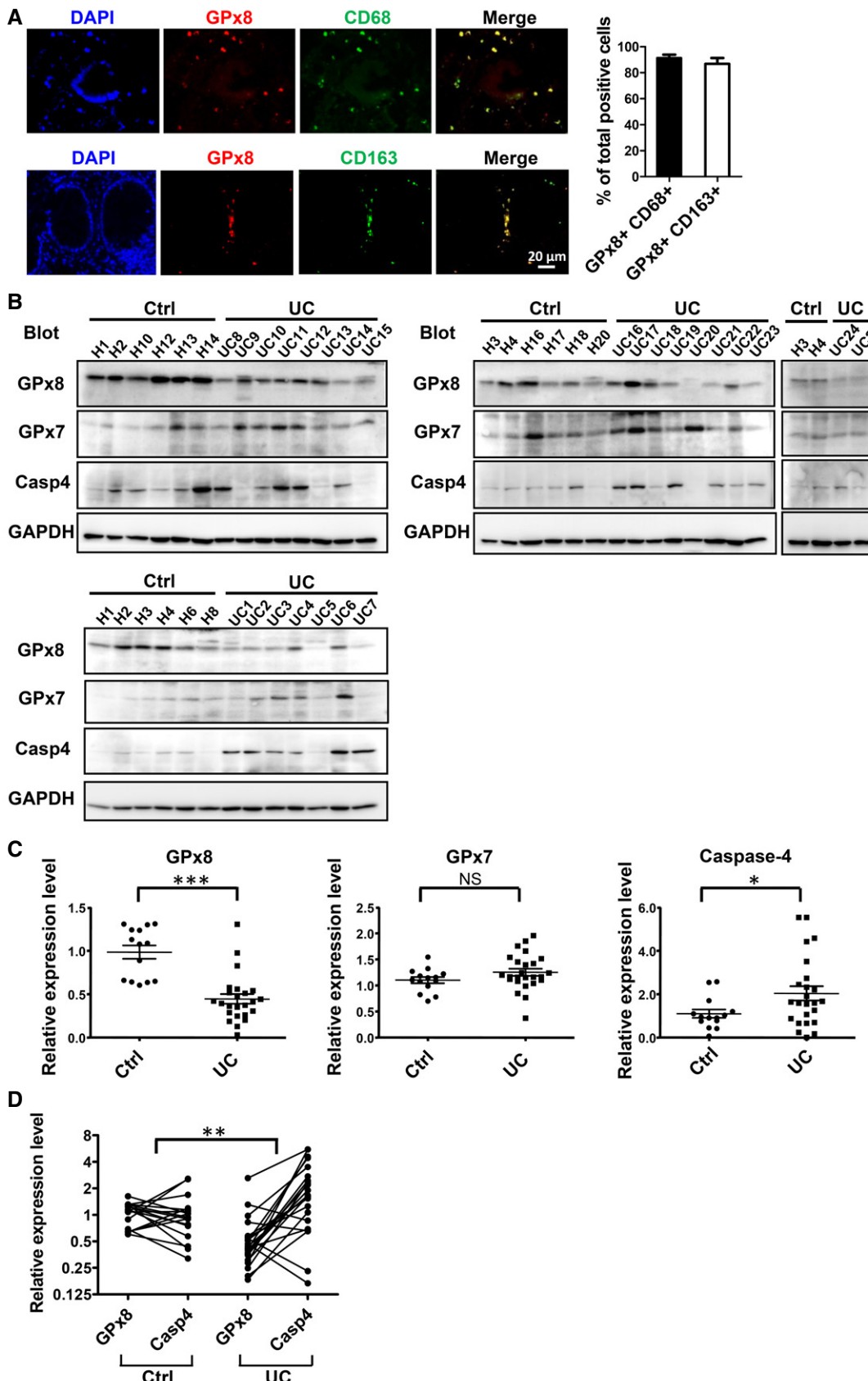

**Figure 7.**

disease manifestation has been reported in UC patients (Flood *et al*, 2015) demonstrating the clinical importance of non-canonical inflammasome signaling in IBD. The downregulation of GPx8 is connected to the overactivation of non-canonical inflammasomes, which can be subdued by a caspase-4/11 inhibitor, VX-765 (Fig 6). Interestingly, VX-765 is a prodrug that requires esterase cleavage to yield its aldehyde functionality. VX-765 is a potent inhibitor of caspase-1 ($K_i$ = 0.8 nM) and caspase-4 ($K_i$ < 0.6 nM) and exhibits at least 100-fold less potency against other subfamily caspases (Wannamaker *et al*, 2007). We consistently found that VX-765 is very effective at modulating caspase-4/11 and non-canonical inflammasome activities, which reduced cytokine production in the colon and reversed morbidity (Fig 6). Preclinical evidence supports the administration of antioxidants with anti-inflammatory activity in the treatment of IBD (Balmus *et al*, 2016). Our data are consistent with this contention, showing that NAC-mediated inhibition of ROS generation prevents the activation of the caspase-4/11-dependent inflammasome pathway and rescues morbidity and mortality in knockout mice (Fig EV4). However, there is no consensus as to the efficacy of NAC in the preclinical treatment of IBD, perhaps due to the complexities of ROS functional mechanisms (Balmus *et al*, 2016). Thus, using a caspase-4/11 inhibitor alone or in combination with an antioxidant may be a promising therapeutic option for IBD patients.

# Materials and Methods

### Plasmids, antibodies, and reagents

*Caspase-4*, *GPx7*, and *GPx8* were PCR-amplified from cDNA constructs (OriGene, Rockville, MD, USA) and cloned into the expression vector pHRSIN (Hsu *et al*, 2015). PCMV-FLAG-caspase-11 and caspase-1 were gifts from Junying Yuan (plasmid #21145, Addgene, Cambridge, MA, USA). Caspase-4 point mutants were constructed using a kit (Quick Change Site-Directed Mutagenesis Kit, Stratagene, San Diego, CA, USA). The N-terminal truncated GPx8 gene was PCR-amplified and cloned into *Escherichia coli* (*E. coli*) protein expression vector pET15. Antibodies used in this study were GPx8 (GeneTex, Irvine, CA, USA; diluted 1:1,000), F4/80 (Bio-Rad, Hercules, CA, USA; 1:200), glyceraldehyde 3-phosphate dehydrogenase (GeneTex; 1:5,000), anti-FLAG epitope (Sigma-Aldrich, Saint Louis, MO, USA; 1:1,000), Myc tag (Genescript, Piscataway, NJ, USA; 1:5,000), caspase-11 (Cell Signaling, Danvers, MA, USA; 1:1,000), caspase-11 (Biolegend, San Diego, CA, 1:1,000), caspase-4 (Cell Signaling; 1:1,000), and caspase-1 (Adipogene, San Diego, CA, USA; 1:1,000), GSDMD (Cell signaling; 1:2,000). LPS, MSU, nigericin, poly(I:C), and Multi-TLR Array™ were obtained from Invivogen. ATP and NAC were obtained from Sigma-Aldrich. VX-765 was obtained from AbMole BioScience (Houston, TX, USA). M-CSF, cytokine kits for mouse and human IL-1β, IL-6 and TNFα, and multiplex immunoassays for detecting cytokines in mouse tissue were obtained from eBioscience.

### Mice

$GPx8^{-/-}$ mice were purchased from Lexicon Genetics, Inc. (Woodlands, TX, USA) (stock number: 032346-ICD). As described by the provider (Tang *et al*, 2010), a $GPx8^{-/-}$ 129S5/SvEvBrd ES cell clone was generated by exchanging exon 1 with a LacZ/Neo selection cassette via homologous recombination. Targeted ES cells were then microinjected into C57BL/6J-$Tyr^{c-Brd}$ blastocysts. The chimeras were backcrossed to C57BL/6J albino mice to generate F1 heterozygous offspring with a germline transmitting a disrupted *GPx8* allele ($GPx8^{+/-}$). To obtain $GPx8^{-/-}$ and WT littermate controls, F1 heterozygotes were intercrossed to produce F2 homozygotes (WT and $GPx8^{-/-}$). Genotyping was performed by polymerase chain reaction (PCR) amplification of mouse tail genomic DNA to differentiate the wild-type (408 bp; 432-5: 5′-GAAGCCGAGAACCAA CAGCTT-3′, 432-6: 5′-CCATAACCCAAATGAATGGTCG-3′) from the *GPx8* knockout allele (218 bp; Neo3a: 5′-GCAGCGCATCGCCTTC TATC-3′, 432-4: 5′-CGATATTACTTTGAAGACTAC-3′). Mice were kept in a pathogen-free environment and were 8–12 weeks of age at the time of the experiments. Age-, sex-, and weight-matched $GPx8^{-/-}$ mice and WT littermates ($GPx8^{+/+}$) were used in comparison studies.

### DSS-induced colitis

Age- and weight-matched 8- to 12-week-old male $GPx8^{-/-}$ mice and WT littermates were co-housed for 3 weeks and used in comparison studies. DSS-induced colitis was induced by the administration of 4% DSS (molecular weight 36,000–50,000, MP Biomedicals, Santa Ana, CA, USA) in the drinking water. DSS drinking water was made available to the mice for 5 or 6 days, followed by 8–9 days of normal water, as previously reported (Zaki *et al*, 2010). In drug treatment groups, $GPx8^{-/-}$ mice received daily intraperitoneal injections of NAC (20 mg/kg) or VX-765 (20 mg/kg), commencing the day before DSS was added to drinking water. Body weight and stool samples were monitored daily, starting from day 0 of treatment, as previously described (Demon *et al*, 2014). On day 14, the colon was removed, measured, photographed, and fixed in 10% formalin. Intestinal inflammation was assessed histologically, using previously described methods (Zaki *et al*, 2010). Animal care and experiments were approved by the Institutional Animal Care and Utilization Committee of Academia Sinica, Taipei, Taiwan.

### Oligomerization analyses by blue native PAGEs

Blue native PAGEs (Invitrogen) were used to analyze LPS-induced protein oligomerization, as previously described (Wittig *et al*, 2006). Purified proteins were incubated with oxidized GPx8 for 30 min at 4°C then incubated with LPS for 30 min at 37°C. The oligomerized proteins were mixed with 1× native gel loading dye and 0.5% Coomassie G-250, then analyzed by 3–12% blue native PAGEs and immunoblotting, according to the manufacturer's instructions (Invitrogen).

### Adoptive cell transfer of BMDMs

After undergoing macrophage depletion by liposomal clodronate, female C57BL/6 mice underwent adoptive cell transfer of BMDMs from female $GPx8^{-/-}$ or littermate control mice, using a previously described protocol (Weisser *et al*, 2012). Mice were supplied with drinking water containing 4% DSS for 5 days, followed by normal water for 2 days. On day 7, guts were collected, measured,

photographed, and fixed in 10% formalin. Intestinal inflammation was assessed histologically, using previously described methods (Weisser et al, 2012).

## Patients and specimens

Human colon specimens were obtained from 25 patients with UC and 20 healthy individuals at the China Medical University in Taichung, Taiwan. Colon tissues were obtained by endoscopy. Informed consent was provided by each subject before entering the study, as according to the requirements issued by the Ethics Committee of China Medical University conforming to the principles set out in the WMA Declaration of Helsinki and the Department of Health and Human Services Belmont Report. The Institutional Review Board approval number for the protocol is CMUH105-REC3-041.

## Macrophage cultures

THP-1 and BMDMs were prepared as previously reported (Hagar et al, 2013; Kayagaki et al, 2015). THP-1 cells were differentiated for 3 days in a culture medium supplemented with 0.1 μM of phorbol 12-myristate 13-acetate (PMA). THP-1 cells were assayed on day 3 in RPMI with 2% serum. BMDMs were isolated from sex-matched mice and differentiated for 5–6 days, following standard procedures as previously described (Kayagaki et al, 2015). Lentiviral transduction of cells was performed as described previously (Hsu et al, 2015). Adherent BMDMs were cultured overnight at a concentration of $1 \times 10^6$ cells per ml before being primed for 5–6 h with the indicated ligand. Primed cells were transfected with LPS using 0.25% v/v Fugene HD (Promega, Madison, WI), as previously reported (Kayagaki et al, 2015). For TLR screening, BMDMs were primed with IFNγ (250 U/ml) (Proteintech, Rosement, IL) and then triggered by each ligand for TLRs for 16 h. A CytoTox 96 non-radioactive cytotoxicity assay (Promega) was used to quantify cell death.

## Purification of recombinant proteins

To obtain N'-truncated GPx8 proteins (N' Δ1-43), E. coli BL-21 (DE3) with the expressing constructs were grown in Luria–Bertani medium supplemented with ampicillin (100 μg/ml). Protein expression was induced with 0.1 mM isopropyl-B-D-thiogalactopyranoside at 25°C for 12 h. To obtain the purified recombinant proteins, bacteria were harvested and proteins were purified by affinity chromatography using Ni-NTA beads (Roche, Basel, Switzerland), as previously described (Hsu et al, 2008). Proteins were dialyzed against a buffer containing 20 mM Tris (pH 8), 150 mM NaCl, and 5 mM 2-mercaptoethanol. Recombinant caspase-4 was overexpressed in 293T and pulled down by FLAG affinity gel (Sigma-Aldrich). After undergoing extensive washing with buffer containing 20 mM Tris (pH 8), 150 mM NaCl, and 1% NP-40, the proteins were eluted with FLAG peptides (Sigma-Aldrich) in 20 mM HEPES buffer and analyzed by SDS–PAGE. To prepare reduced or oxidized GPx8, the proteins were treated with 1,4-dithiothreitol (DTT, Sigma-Aldrich) or $H_2O_2$, respectively, for 30 min. DTT and $H_2O_2$ were removed by buffer exchange using Amicon Ultra-10 (Millipore, Billerica, MA, USA). Oxidized or reduced proteins were used in the assays immediately after preparation.

## Gut microbiota analysis

To obtain the $GPx8^{-/-}$ and WT littermate controls, F1 heterozygotes were intercrossed to produce F2 homozygotes (WT and $GPx8^{-/-}$) and used for further F3 mice generation. Feces from F3 mice of mixed gender $GPx8^{-/-}$ and littermates were subjected to gut microbiota analysis. Feces were collected and stored at −80°C before undergoing DNA isolation, using a kit according to the manufacturer's instructions (QIAamp DNA Microbiome Kit). Microbial DNA was subjected to 16S rDNA amplicon sequencing using Illumina sequencing-by-synthesis technology to produce $2 \times 300$ bp paired-end reads. 16S metagenomics analysis including taxonomic composition, α and β analysis, was performed by Welgene Biotech Co., Ltd.

## Co-immunoprecipitation (Co-IP) assays

Whole-cell lysates were prepared using a lysis buffer containing 20 mM Tris–Cl (pH 8), 1% Triton X-100, 150 mM NaCl, and a protease and phosphatase inhibitor cocktail (Roche), followed by 20 min of centrifugation at 13,000 g at 4°C. For Co-IP, around 500 μg of the crude whole-cell extract was pre-cleaned with beads at 4°C for 1 h and then incubated with 1 μg of antibodies or control IgG antibodies at 4°C for 3 h. After washing with lysis buffer (1% Triton X-100), the interacting proteins were eluted with SDS buffer with or without DTT and analyzed by immunoblotting. Intensity of proteins was quantified by ImageJ software.

## Streptavidin pull-down assay

To assay LPS binding to FLAG-tagged caspase-4 in the presence of GPx8, biotin-conjugated LPS (Invivogen, Carlsbad, CA, USA) was immobilized on streptavidin beads (Roche). Unconjugated LPS was removed by washing with assay buffer containing 20 mM HEPES (pH 7.4), 150 mM NaCl, and 0.01% Tween-20. The beads were co-incubated with 0.1 μg caspase-4 pretreated with oxidized GPx8 for 30 min at 37°C. Unbound proteins were washed with assay buffer and analyzed by immunoblotting.

## Endotoxic shock model

8- to 12-week-old female mice were primed with ultra-pure LPS (E. coli O55:B5, 400 μg/kg) then re-challenged after 6 h with LPS (40 mg/kg) by intraperitoneal injections. Alternatively, mice were primed with poly(I:C) (LMW 1 mg/kg) then re-challenged 6 h later with intraperitoneal LPS (10 mg/kg). IL-1β production levels in mouse sera were measured 2 h after mice were primed with poly(I:C) (LMW 1 mg/kg) then re-challenged after 6 h with LPS (1 mg/kg).

## Quantitative reverse transcriptase PCR

Total RNA was extracted from cells using TRIzol® (Life Technologies), according to the manufacturer's instructions. RNA was reverse-transcribed into cDNA using commercially available kits (Roche). PCRs were performed using the Applied Biosystems 7300 Real-Time system (Applied Biosystems) incorporating SYBR Green

**The paper explained**

**Problem**

Human caspase-4 and its mouse homolog caspase-11 are receptors for cytoplasmic lipopolysaccharide. Activation of the caspase-4/11-dependent NLRP3 inflammasome is required for innate defense and endotoxic shock, but how caspase-4/11 is modulated remains unclear.

**Results**

In this study, we show that mice lacking the oxidative stress sensor GPx8 are more susceptible to colitis and endotoxic shock, and exhibit reduced richness and diversity of the gut microbiome. C57BL/6 mice that underwent adoptive cell transfer of GPx8-deficient macrophages displayed a similar phenotype of enhanced colitis, indicating a critical role of GPx8 in macrophages. GPx8 binds covalently to caspase-4/11 via disulfide bonding between cysteine 79 of GPx8 and cysteine 118 of caspase-4, and thus restrains caspase-4/11 activation, while GPx8 deficiency leads to caspase-4/11-induced inflammation during colitis and septic shock. Inhibition of caspase-4/11 activation with either N-acetylcysteine, a strong antioxidant, or VX-765, a caspase-4 inhibitor, reduces colitis in Gpx8-deficient mice. Notably, colonic tissues from patients with ulcerative colitis display low levels of Gpx8 and high caspase-4 expression.

**Impact**

These results suggest that GPx8 protects against colitis by negatively regulating caspase-4/11 activity. GPx8 appears to play a negative regulating role in the non-canonical inflammasome pathway in response to cellular ROS levels, implying that GPx8 has impacts on the pathogenesis of IBD and relevant disease pathways. Thus, using a caspase-4/11 inhibitor alone or in combination with an antioxidant may be a promising therapeutic option for IBD patients.

dye (Roche). The threshold cycles of all triplicates were normalized to GAPDH. All primers used are listed as follows: mouse *GPx8* primers, 5′-TCCTGAAGCCGAGAACCAAC and 5′-CTGTGAAGCGG CAGTCACTA; human *GPx8* primers, 5′-CCTCAAGAATGCCAGATG and 5′-CACCTTAGTAAGTGTGTTAATTG; *caspase-11*, 5′-TGTCTTCA CGGTGCGAAAGA and 5′-CAGGGTGTTTGTTTTCAGCCA.

**Immunofluorescence staining**

THP-1 cells were differentiated for 3 days in a culture medium supplemented with 0.1 μM of phorbol 12-myristate 13-acetate (PMA). Cells seeded in 12-well plates were primed for 5–6 h with the indicated ligand. Primed cells were transfected with 2.5 μg LPS by using 0.25% v/v Fugene HD (Promega, Madison, WI, USA). Cells were washed twice, fixed in 4% paraformaldehyde, and permeabilized with 0.1% Triton X-100. The cells were incubated with anti-Myc (Genescript; diluted 1:500), FLAG (Cell Signaling; 1:1,000), GPx8 (GeneTex; 1:50), CD68 (Biolegend, USA; 1:100), CD163 (Leica Biosystems, Buffalo Grove, IL, USA; 1:200), CD103 (Leica Biosystems; 1:100), F4/80 (Bio-Rad; 1:50), and anti-caspase-4 (Cell Signaling; 1:100) and then with either Alexa 488-labeled or Alexa 647-labeled antibody to rabbit or mouse IgG (Invitrogen; 1:300). Nuclei were stained with DAPI (4′,6-diamidino-2-phenylindole, Dojindo, Rockville, MD, USA). Co-localization was quantified by Pearson's correlation coefficient by ImageJ software, in order to measure the strength of co-localization between proteins; the formulas return a value between −1 and 1, where 1 indicates a strong positive relationship.

**In situ proximity ligation assays**

Fixed and permeabilized cells were incubated overnight at 4°C with primary antibodies: rabbit anti-caspase-4 (Cell Signaling; diluted 1:100) and mouse to Myc (Genescript; 1:500). The cells were washed and allowed to react to a pair of proximity probes (Sigma-Aldrich). The remainder of the *in situ* PLA protocol was performed according to the manufacturer's instructions. The cells were examined using confocal microscopy.

**Statistics**

Results are presented as the means ± SEM in mouse model experiments and clinical sample analysis and as the means ± SD in experiments using primary macrophages or THP-1 cells. Mice were allocated to experimental groups on the basis of their genotype and randomized within the given group to a treatment group. For the *in vivo* experiments, the investigator was usually blinded toward the genotype, but not the treatment group. Sex-matched mice were included in equal numbers for each animal experiment. All statistical analyses were performed using GraphPad Prism software. Data were analyzed using the Student's *t*-test or Mann Whitney test for non-normally distributed data. Mice survival data were analyzed by the Mantel–Cox method. The *P*-value of GPx8 and caspase-4 expression paired in each individual was analyzed by Fisher's exact test. Statistical analyses and replicate numbers for each experiment are indicated in figure legends. A *P*-value of < 0.05 was considered significant. NS: not significant ($P > 0.05$). *$P < 0.05$; **$P < 0.01$; *** $P < 0.001$. While figure legends only show *P*-value ranges, exact *P*-values can be found in the corresponding Appendix Table S1.

**Expanded View** for this article is available online.

## Acknowledgements

This work was supported by grants from the Academia Sinica, Taiwan Peak Project (grant numbers 2371 and 4012), the Ministry of Science and Technology (Most grant number 105-2628-B-039-003-MY3, 104-2320-B-039-050, and 108-2320-B-039-037), China Medical University (grant numbers CMU106-N-14), and the Ministry of Education (Sprouting Project), Taiwan. We thank the National Laboratory Animal Center (NLAC), NARLabs, Taiwan, for their support with animal care and breeding, and the National RNAi Core Facility at Academia Sinica in Taiwan for providing shRNA reagents and related services. We thank Dr. Shie-Liang Hsieh for the critical comments, and Dr. Ping-Ning Hsu for the assistance in this study. We also thank Iona J. MacDonald for editing the manuscript.

## Author contributions

J-LH and W-HL designed the experiments. J-LH and T-FC conducted the experiments. F-YS conducted supplementary experiment 1. J-LH, J-TH, J-WC, J-LL, and P-CW analyzed the data. J-LH and W-HL wrote the paper. J-WC, J-LL, P-CW. C-MH, and EY-HPL edited and commented on the manuscript.

## Conflict of interest

The authors declare that they have no conflict of interest.

## For more information

(i)  https://www.uniprot.org/uniprot/Q8TED1
(ii)  https://www.uniprot.org/uniprot/P49662

(iii) https://www.uniprot.org/uniprot/P70343
(iv) http://www.genomics.sinica.edu.tw/index.php/en/lee-wen-hwa

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
