## [Review Process File · EMBO Molecular Medicine]

Glutathione peroxidase 8 negatively regulates caspase-4/11 to protect against colitis

Jye-Lin Hsu, Jen-Wei Chou, Tzu-Fan Chen, Jeh-Ting Hsu, Fang-Yi Su, Joung-Liang Lan, Po-Chang Wu, Chun-Mei Hu, Eva Y-HP Lee, and Wen-Hwa Lee

Review timeline:	Submission date:	15 February 2019
	Editorial Decision:	15 March 2019
	Resubmission received:	27 June 2019
	Editorial Decision:	22 July 2019
	Revision received:	13 September 2019
	Editorial Decision:	17 October 2019
	Revision received:	24 October 2019
	Accepted:	30 October 2019

Editor: Céline Carret

Transaction Report:

1st Editorial Decision

15 March 2019

Thank you for the submission of your manuscript "Glutathione peroxidase 8 negatively regulates caspase-4/11 to protect against colitis". We have now heard back from the two referees whom we asked to evaluate your manuscript.

As you will see, the reviewers raised a number of serious conceptual flaws and experimental shortcomings of the study, and feel that given these limitations, your conclusions appear to not be fully supported by the data. As clear and conclusive insight into a novel clinically relevant observation is key for publication in EMBO Molecular Medicine, and together with the fact that we only accept papers that receive enthusiastic support upon initial review, I am afraid that we cannot offer to consider the manuscript further.

We hope that the referee comments will be helpful to you as you prepare your manuscript for submission elsewhere. Thank you again for your interest and we hope that you will continue to consider sending your work to EMBO Molecular Medicine in the future.

***** Reviewer's comments *****

Referee #1 (Remarks for Author):

In this manuscript, the authors describe a new knockout mouse model for glutathione peroxidase 8 (GPX8) and propose that loss of GPX8 aggravates DSS-induced colitis and LPS-induced sepsis. The authors go on and propose that by interfering with caspase-4/11 activation GPX8 negatively regulates noncanonical inflammasome activation. While the concept would be of interest, there are a series of experimental flaws and technical shortcomings that limit the strength of the study.

Moreover, many experiments show only very subtle differences, so I wonder whether there is really any relevant meaning to them.

Main concerns:

- The authors purchased mice from Lexicon Genetics where a lacZ/neo cassette was inserted into exon 1 of the Gpx8 gene. The authors failed to carefully characterize the actual knockout of the Gpx8 gene and just provided an immunoblot in BDMDs using an antibody made against human GPX8 (suppl. Figures 1&2). Thus, the authors need to fully characterize the knockout in different tissues using additional tools such as RT-PCR, more reliable antibodies etc in order to rule out that truncated products/splice variants are being generated in these mice. Notably, exon 2 harbors the active site C79! Without a thorough and careful examination of these mice data generated in the different models are obsolete and hard to judge.
- Figure 1: Besides the limitations mentioned above, can the authors reliably exclude that not different microbiota between knockout mice and control groups (present at the time of purchase) may cause the observed differences in the DSS-induced colitis model as the gut microbiota itself can have profound effects in the same background strains (see Li et al 2018). Did the authors use littermates coming from heterozygous breeding?
- Figure 2: the authors mention mortality in the main text but this is not shown. Additionally, the differences shown in 2C/2D are very subtle.
- Figure 3: Again, many of the differences shown are subtle (A, B, D). Did the authors try to re-express wt GPX8 to blunt see whether these effects can be blunted?
- Fig 3C: wrong labeling?
- Fig 4A: It is hard to follow what actually has been done. A much improved labeling/description of the experiments done is mandatory (which applies not only to this but many other experiments in the manuscript!). The lack of staining against FLAG is confusing.
- Fig. 4B: The quality is poor. Moreover, GPX8 is a type-II transmembrane ER-resident protein. Is caspase-4/11 located to the same compartment? That being said, it is very hard to identify "specific" interacting partners of GPX8 as it sits in the membrane with a highly reactive Cys in its active site, leading to many false-positive results. "Co-localization" could only be detected in cells, where Myc (GPX8) is overexpressed.
- Fig. 4D: the complexes shown are less surprising as GPX8 in the absence of reducing compounds will (like all other GPX) form complexes.
- Fig4G and F: the proteins produced are they are actually properly folded?
- Fig. 5A/B: again subtle differences questioning the relevance of the findings.
- Fig. 6E: In contrast to Fig 3D, there is no difference in Caspase-1 activation between the WT and KO groups - VX-765.

Referee #2 (Remarks for Author):

The authors analyzed the contribution of GPx8 in colitis and in controlling the induction of colitis. The authors show that mice KO for GPx8 have increased expression of IL1b and activation of the inflammasome. Then the authors show a binding of GPx8 with caspase 4/11 and conclude that Gpx8 is involved in controlling the activation of the non canonical pathway of the inflammasome activation.

While I found the results interesting, I am not sure that the authors have really demonstrated that GPx8 really acts through the non canonical pathway.

Major points

Fig. 1. 4% DSS is a very high amount of DSS especially on mice with C57/BL6 background. No clear how many times was the experiment repeated.

The western blot in figure S2B for GPX8 is not very clear. The specific band seems to be very low. What is the production of other cytokines like TNF α and IL-10 that are normally induced after colitis?

Fig. 2 The role of macrophages is not really clear in this figure, maybe also due to the experimental model as one expects that clodronate dependent depletion of macrophages does not last long and this may explain why the experiment was stopped at 7 days with a minor effect on body weight loss. Do the authors have an explanation for the increase of macrophages in the absence of GPX8, in figure 2G. Why is the number of samples drastically reduced in Fig. 2H?

In addition, the authors have not shown mortality which is reported in the text or activated macrophages. The whole of figure 2 description in the text does not correspond to the actual figure.

Fig. 3. It was clear from the previous figures that GPX8 affected the inflammasome as it induced IL1b release. LDH is a marker of cell lysis, not specifically of pyroptosis. Caspase 1 also seems to be strongly induced (even before caspase 11 activation) suggesting both a classical and non classical activation of the inflammasome. The presence of pyroptosis should converge towards a classical activation rather than non classical.

Fig. 4 It is very easy to have a false positive result in co-immunoprecipitation. In the blots of 4A, the co-ip seem to be very weak and this is confirmed in the colocalization experiment in 4B. Here a quantification of the colocalization should be shown. Further it is not clear whether the cells that are shown have been activated with LPS (and pretreatment) and if so what is the level of colocalization of untreated cells? Why in figure 4C only one treatment with LPS was performed and not the pretreatment, or was it a transfection? I have the impression that GPX8 major target is the classical activation of the inflammasome. Why haven't the authors analyzed also the binding to caspase 1? The finding that the mutants abolish the inhibitory activity does not imply that this occurs through caspase4 interaction.

Fig. 6. The experiments with the casp1/4 inhibitor VX-765 are very interesting but do not rule out the possibility that GPX8 actually does not interact through caspase 4..

Fig 7A the control of uninflamed area is missing. What is the expression of caspase 1 and activated gasdermin D? While the downregulation of GPx8 is clear, the upregulation of caspase 4 is not, further why haven't the authors analyzed the activated form of caspase 4?

Resubmission - authors' response

27 June 2019

[Authors have resubmitted their article without being invited to do so. After reading the rebuttal letter and thoroughly evaluating the revised article, editors decided to move forward with the submission and sent the revised article back to the referees].

We appreciate that both reviewers consider the novelty of the finding and thank for all the comments that help us improve the strength of this paper. The main concerns are whether knockout mice are truly deficient of GPx8 and whether GPx8 regulates only caspase-4/11 mediated noncanonical inflammasome. We have fully addressed these two questions and all other comments. We demonstrated that either GPx8 protein or RNA is not detected in knockout mice. We also showed that GPx8 does not interact with caspase-1 and has no role in directly modulating canonical inflammasome via caspase-1. We answer all the comments in point-by-point details as attached below:

Detailed point-by-point response to reviewers' comments
All Referee comments are in blue and answers are in black.
Referee #1 (Remarks for Author):

In this manuscript, the authors describe a new knockout mouse model for glutathione peroxidase 8 (GPX8) and propose that loss of GPX8 aggravates DSS-induced colitis and LPS-induced sepsis. The authors go on and propose that by interfering with caspase-4/11 activation GPX8 negatively regulates noncanonical inflammasome activation. While the concept would be of interest, there are a series of experimental flaws and technical shortcomings that limit the strength of the study. Moreover, many experiments show only very subtle differences so I wonder whether there is really any relevant meaning to them.

Response: Thank you for believing that the paper is interesting. We shall clear up all the concerns raised.

Main concerns:

1. The authors purchased mice from Lexicon Genetics where a lacZ/neo cassette was inserted into exon 1 of the Gpx8 gene. The authors failed to carefully characterize the actual knockout of the Gpx8 gene and just provided an immunoblot in BDMDs using an antibody made against human GPX8 (suppl. Figures 1&2). Thus, the authors need to fully characterize the knockout in different tissues using additional tools such as RT-PCR, more reliable antibodies etc in order to rule out that

truncated products/splice variants are being generated in these mice. Notably, exon 2 harbors the active site C79! Without a thorough and careful examination of these mice data generated in the different models are obsolete and hard to judge.

Response: Following the suggestions of reviewer 1, we have clearly verified that our *GPx8*^{-/-} mice are defective in GPx8 protein and RNA expression. We carefully demonstrated GPx8 expression in different organs of WT and *GPx8* knockout mice by using immunoblots. The antibody from GeneTex was generated by immunization rabbit with purified full-length recombinant GPx8 protein provided by us. We can see a specific band in tissues from wild type but not *GPx8*^{-/-} mice. No specific band was detected in *GPx8*^{-/-} mice, suggesting no short form or truncated product was produced in *GPx8*^{-/-} mice. In addition, we also designed reverse transcription PCR primers amplified the region (129-241bp of GPx8) cross exon 1 and 2 junction showed that there is no GPx8 RNA expression in *GPx8*^{-/-} cells (R. Fig. 1). We have incorporated R. Fig. 1 into the Revised Supplementary Fig. 1.

R. Fig. 1 *GPx8*^{-/-} mice are defective in GPx8 protein and RNA expression. (A) GPx8 expression in various organs from WT or *GPx8* knockout mice (KO) mice by immunoblots with antiGPx8 antibody. (B) Relative mRNA levels of *GPx8* in WT and *GPx8*^{-/-} BMDMs.

2. Figure 1: Besides the limitations mentioned above, can the authors reliably exclude that not different microbiota between knockout mice and control groups (present at the time of purchase) may cause the observed differences in the DSS-induced colitis model as the gut microbiota itself can have profound effects in the same background strains (see Li et al 2018). Did the authors use littermates coming from heterozygous breeding?

Response: Thank you for the suggestions. We have considered microbiota be a factor for colitis model. Therefore, we used not only littermate controls but also co-housed mice in our colitis model, described in material and methods to minimize this factor as previously described (Demon, Kuchmiy et al., 2014). In addition, we further confirmed the phenotypes by adoptive transfer macrophages to B6 mice to exclude microbiota interference described latter in the letter.

3. Figure 2: the authors mention mortality in the main text but this is not shown. Additionally, the differences shown in 2C/2D are very subtle.

Response: Due to the experimental limitation, the inflammation phenotypes was examined in 7 days, instead of 14 days, period with moderate body weight loss and clinical score changes but without mortality. In this condition of short experiment period, no mortality change was observed. As reviewer 2 mentioned, in the experimental model we used, clodronate dependent depletion of macrophages does not last long because renewed macrophages will come back around 10-14 days (Bader, Enos et al., 2018, Weisser, van Rooijen et al., 2012). Thus, we did not observe mortality under this condition. However, this adoptive transfer model can demonstrate the effect of GPx8-depleted macrophages in the same background of mice with the same microbiota. The differences shown in 2C/2D were similar to 1C/1D at the 7-day period. In addition, the over-production of inflammatory cytokines and clinical phenotypes were observed in animals transplanted with *GPx8*^{-/-} macrophages, suggesting the important function of GPx8 in macrophages.

4. Figure 3: Again, many of the differences shown are subtle (A, B, D). Did the authors try to re-express wt GPX8 to blunt see whether these effects can be blunted?

Response: Thank you for the comments. We did the rescue experiment as reviewer suggested by overexpressing human Gpx8 in BMDMs via lentiviral transduction for 1 week and analyzed cell

deaths and IL-1 β production after LPS transfection. RNA expression of human GPx8 was confirmed by quantitative PCR (R Fig.2 below). These results showed that expressing human GPx8 (hGPx8) blunted the phenotypes of *Gpx8*^{-/-} BMDMs. We have incorporated R. Fig. 2 into the Revised Fig. 3.

R. Fig. 2 Over-activation of noncanonical inflammasome induced by GPx8-deficiency in BMDMs can be rescued by ectopically expressing human GPx8 (hGPx8). (A) The activation of noncanonical inflammasome in BMDMs with empty vector (EV) or hGPx8 was verified by IL-1 β and LDH release. (B) Expression of hGPx8 mRNA in BMDMs was confirmed by quantitative reverse transcriptase PCR.

5 Fig 3C: wrong labeling?

Response: We corrected the label.

6 Fig 4A: It is hard to follow what actually has been done. A much improved labeling/description of the experiments done is mandatory (which applies not only to this but many other experiments in the manuscript!). The lack of staining against FLAG is confusing.

Response: Thank you for the suggestions. We clarified descriptions in figure legends and improved intensity of immunoblots for input controls (R. Fig. 3) by a longer exposure time.

R. Fig. 3 Interaction of GPx8 with human caspase-4 (Casp4) or mouse caspase-11 (Casp11) by co-IP assays. 293T cells were co-transfected with vector expressing FLAG tagged Casp4 or Casp11, along with GPx8 expressing vector. (A) 293T cell lysates containing Casp4- or Casp11-FLAG proteins were immunoprecipitated by anti-FLAG Ab and subsequently analyzed by Western blots using anti-GPx8 and anti-FLAG Ab. (B) GPx8 and its interacting proteins were precipitated by anti-GPx8 Ab and analyzed by indicated Abs.

7. Fig. 4B: The quality is poor. Moreover, GPX8 is a type-II transmembrane ER-resident protein. Is caspase-4/11 located to the same compartment? That being said, it is very hard to identify "specific" interacting partners of GPX8 as it sits in the membrane with a highly reactive Cys in its active site, leading to many false-positive results. "Co-localization" could only be detected in cells, where Myc (GPX8) is overexpressed.

Response: Thanks for the suggestion. After zoomed in with the original picture, Co-localization of GPx8 and caspase-4 was clearly observed in the same compartment in THP1 cells (R. Fig 4A). Reciprocally, the co-localization can be detected when Casp4-Flag was expressed (R. Fig.4B). We have incorporated R. Fig. 4 into the Revised Fig. 3 and Fig. S4.

R. Fig. 4 Co-localization of GPx8 and caspase-4 by immunofluorescence assays. (A) Co-localization of endogenous caspase-4 with Myc tagged GPx8 (GPx8-Myc). (B) Co-localization of endogenous GPx8 with FLAG tagged caspase-4 (Casp4-FLAG).

8 Fig. 4D: the complexes shown are less surprising as GPX8 in the absence of reducing compounds will (like all other GPX) form complexes.

Response: Thank you for the suggestions. Due to missing experimental details, the reviewer may overlook this point. We performed co-immunoprecipitation assay with anti-FLAG Ab to immunoprecipitate caspase-4(C258S) containing complexes. We further revealed that these high molecule complexes are GPx8-caspase-4 complexes but not GPx8-GPx8 shown in R. Fig. 5. We performed the experiment again and removed unnecessary figures to emphasize this point.

R. Fig. 5 Complexes of GPx8 and caspase-4 were demonstrated by co-IP assays. 293T cells were co-transfected with vectors expressing FLAG tagged caspase-4 (C258S), along with GPx8 expressing vectors. Cell lysate was immunoprecipitated by anti-FLAG Ab and subsequently analyzed by Western blots using anti-GPx8 or anti-FLAG Ab. Cysteine-dependent covalent interactions of GPx8 and caspase-4 were analyzed under non-reducing and reducing conditions. Disulfide-linked complexes of Gpx8 and caspase-4 are indicated.

9. Fig4G and F: the proteins produced are they are actually properly folded?

Response: GPx8 mutant proteins in Fig4G and 4F expressed in stable cell lines are most likely to be properly folded. Since unfolded proteins are usually degraded by proteasome or accumulated as insoluble aggregates in cells. GPx8C79S and C108S are normally expressed with comparable amount of WT protein in cells and can be detected as soluble proteins, inferred they are properly folded. GPx8C2S2 expressed less in the cells, likely some partial unfolded proteins were degraded, but some of it can be detected in soluble form. Although it is nearly impossible to be sure that these proteins are actually properly folded, these GPx8 mutants detected in soluble forms with compatible amount are likely properly folded in cells.

10 Fig. 5A/B: again subtle differences questioning the relevance of the findings.

Response: To clarify this point, we quantified complexes formation between GPx8 and Caspase 4 C118S in Fig. 5A (R. Fig. 6A), and showed that complexes formation was reduced to 30% in both reducing and non-reducing conditions. No doubt, this is a significant difference. Furthermore, when we performed the functional assay with this mutant in cells, the inhibition by GPx8 to this mutant was abolished (Revised Fig5C and D), suggesting the importance of C118 of Caspase 4 for the

interaction with GPx8. Original Fig 5B (see also R. Fig. 6B) is an experiment showing mutant proteins expressed equally in THP-1 cells before we did functional assays. The reviewer may overlook this point.

R. Fig. 6 GPx8 inhibits the activity of caspase-4 by covalently binding to cysteine 118. (A) Casp4C118S mutant partially abolished the covalent disulfide-link with GPx8. Interacting Gpx8 were co-immunoprecipitated with FLAG-tagged caspase-4 (WT, C118S, C329S mutants) and analyzed under reducing (+ DTT) or non-reducing (- DTT) conditions. Disulfide-linked complexes of Gpx8 and caspase-4 are indicated. Fold change of complexes was normalized with internal controls and indicated under blots. (B) THP-1 cells stably expressing empty vector (EV), FLAG-tagged caspase-4 (WT, C258S, C118S mutants), or GPx8 (WT and C79S mutants) were confirmed by immunoblotting.

11. Fig. 6E: In contrast to Fig 3D, there is no difference in Caspase-1 activation between the WT and KO groups - VX-765.

Response: In original Fig 3D, we can detect caspase 1 over-activation. In Fig 6E, due to the overexposure, the activated caspase 1 was blurred in the western blot analysis. We reanalyzed the samples with carefully quantification as shown in R. Fig. 7, caspase 1 was over-activated in colon tissues of *GPx8*^{-/-} mice. In addition, active forms of caspase 11 (p26) and GSDMD (p30) were both increased in *GPx8*^{-/-} mice, which is consistent with original Fig. 3D.

R. Fig. 7 Western blot analysis of colon tissues of mice treated with Caspase-4/11 inhibitor, VX-765. Full-length and cleavage forms of caspase-11, caspase-1, and GSDMD were analyzed by immunoblots on day 7 after DSS treatment.

Author):

The authors analyzed the contribution of GPx8 in colitis and in controlling the induction of colitis. The authors show that mice KO for GPx8 have increased expression of IL1b and activation of the

Referee #2 (Remarks for

inflammasome. Then the authors show a binding of GPx8 with caspase 4/11 and conclude that Gpx8 is involved in controlling the activation of the non canonical pathway of the inflammasome activation.

While I found the results interesting, I am not sure that the authors have really demonstrated that GPx8 really acts through the non canonical pathway.

Response: Thank you for the reviewer believing that the story is interesting. We hope that our data have unambiguously demonstrated that Gpx8 regulates caspase-4/11 dependent pathway by modulating capase-4/11 activity. First, GPx8-deficiency enhances caspase-4/11 activation only in the condition of transfection of LPS, which is a caspase-4/11 specific stimulator. It has been shown that in the absence of caspase-11, caspase-1 activation doesn't occur in LPS transfection (Man, Karki et al., 2017, Shi, Zhao et al., 2014). Since direct activation of caspase-4 induces cell death and cleavage of caspase-4/11, both phenotypes were increased in *GPx8*^{-/-} cells and mice. Second, other activators known to activate NLRP3 inflammasomes showed no differential activation in *GPx8*^{-/-} cells (R. Fig. 8). Third, GPx8 can form disulfide bonding complexes with capse-4/11 but not caspase-1 (R. Fig. 9). The critical residue C118 of caspase-4/11, for interacting with GPx8 is not conserved in caspase-1. This interaction can alleviate caspase-4 activation and oligomerization ability *in vitro*. Finally, expressing GPx8 in THP-1 cells, but not mutants abolished the disulfide bonding ability, suppressed caspase-4 activation in cells. Based on those data, it is clear that GPx8 regulates caspase-4/11 dependent pathway, but not caspase-1 activity.

Major points

1 Fig. 1. 4% DSS is a very high amount of DSS especially on mice with C57/BL6 background. No clear how many times was the experiment repeated.

Response: In figure 1, there are two conditions of DSS treatment, one lethal does and one nonlethal does. Totally, we repeated two times of each dose and used 5-6 mice per group. We combined two lethal dose experiments for Fig. B, C and D. For Fig. E, F, and G, we used nonlethal does and combined two nonlethal experiments for Fig. H.

2 The western blot in figure S2B for GPX8 is not very clear. The specific band seems to be very low.

Response: We reanalyzed the GPx8 expression and found that the protein size is correct (R Fig.8). The specific band is the correct size compared with markers (R Fig.8). We have incorporated R. Fig. 8 into the Revised Supplementary Fig. 1.

R. Fig. 8 GPx8 is expressed predominantly in macrophages and not in T, B, or dendritic cells. Cell-type specific expression of GPx8 was elucidated by immunoblotting. Black arrowhead denotes GPx8. White arrowhead denotes non-specific bands. DCs, dendritic cells.

3 What is the production of other cytokines like TNF α and IL-10 that are normally induced after colitis?

Response: As reviewer suggested, we measured TNF- α and IL-10 in our model. Similar to IL-6 and IL-1 β , TNF- α and RANTES productions in *GPx8*^{-/-} mice were enhanced in colitis condition (R Fig. 9B). In addition to that, the numbers of the infiltrated T cells and the amount of secreted IFN γ were also higher in GPx8-deficient mice compared with the WT mice (R Fig. 9A and B), suggesting adaptive immune response was induced after over-production of pro-inflammatory cytokines in mice. However, anti-inflammatory cytokine, such as IL-10, was not changed (R Fig. 9B).

R. Fig. 9 (A) Infiltrated T cells and productions of cytokines in colon were demonstrated on day 14 after DSS treatment. Colon sections were stained with markers for T cells (CD3). (B) Productions of cytokines in colon tissue lysates were analyzed by ELISA. Data are presented as the mean \pm SEM. NS: not significant ($P > 0.05$). * $P < 0.05$; ** $P < 0.01$ (unpaired Student's *t*-test).

4-1 Fig. 2 The role of macrophages is not really clear in this figure, maybe also due to the experimental model as one expects that clodronate dependent depletion of macrophages does not last long and this may explain why the experiment was stopped at 7 days with a minor effect on body weight loss.

Response: Indeed, due to the experimental limitation, the inflammation phenotypes was examined in 7 days, instead of 14 days, period with moderate body weight loss and clinical score changes but without mortality. In this condition of short experiment period, no mortality change was observed. As the reviewer mentioned, in the experimental model we used, clodronate dependent depletion of macrophages does not last long because renewed macrophages will come back around 10-14 days (Bader, Enos et al., 2018, Weisser, van Rooijen et al., 2012). Thus, we did not observe mortality under this condition. However, this adoptive transfer model can demonstrate the effect of GPx8-depleted macrophages in the same background of mice with the same microbiota. The differences shown in 2C/2D were similar to 1C/1D at the 7-day period. In addition, the over-production of inflammatory cytokines and clinical phenotypes were observed in animals transplanted with GPx8^{-/-} macrophages, suggesting the important function of GPx8 in macrophages.

4-2 Do the authors have an explanation for the increase of macrophages in the absence of GPX8, in figure 2G.

Response: In the absence of GPx8, over-activation of noncanonical inflammasome may lead to over-producing of IL-1 β , and downstream pro-inflammatory cytokines. IL-6, TNF- α , and RANTES productions were also more in the deficiency of GPx8. The chemokines, such as RANTES, could recruit macrophages to the inflamed sites and showed in Fig 2G, which is consistent with Revised Fig. 1G.

4-3 Why is the number of samples drastically reduced in Fig. 2H?

Response: In Fig 2, we performed 2 repeats of experiments, and used 9 mice in the first time and 6 mice in the second time. Because we collected tissues for Fig. 2H from the second time, the number of mice are not the same.

4-4 In addition, the authors have not shown mortality, which is reported in the text or activated macrophages. The whole of figure 2 description in the text does not correspond to the actual figure.

Response: Thank you for the suggestion. We have clarified the description for figure 2.

5-1 Fig. 3. It was clear from the previous figures that GPX8 affected the inflammasome as it induced IL1 β release. Caspase 1 also seems to be strongly induced (even before caspase 11 activation) suggesting both a classical and non classical activation of the inflammasome.

Response: To clarify whether GPx8 could affect classical inflammasome activation, we performed functional assays measuring canonical inflammasome activation. We did not observe any differential phenotype under multiple canonical inflammasome stimuli in different time points (R. Fig. 10). In addition, we also checked whether GPx8 might physically interact with caspase-1. We demonstrated that GPx8 cannot interact with caspase-1 by co-IP experiment (R. Fig. 11). It is a misunderstanding that caspase 1 is strongly induced as the reviewer mentioned “even before caspase 11 activation”. Very little difference in pro-caspase-1 protein level was observed before LPS transfection (Revised Fig. 3F, lane 1 and 6). We have incorporated R. Fig. 9 into the Revised Supplementary Fig. 3.

R. Fig. 10 Canonical NLRP3 inflammasome activation was not altered by GPx8 deficiency. BMDMs were primed for 6 h with 0.5 μ g/ml LPS and then stimulated with ATP (5 mM), MSU (150 μ g/ml), or nigericin (5 μ M) for indicated time.

R. Fig. 11 GPx8 does not interact with caspase-1. Interaction of GPx8 with human caspase-4C258S (Casp4) or human caspase-1 (Casp1) was confirmed by co-IP assays. 293T cells were co-transfected with vector expressing FLAG tagged Casp4C258S or Casp1, along with GPx8 expressing vector. 293T cell lysates

containing Casp4- or Casp1-FLAG proteins were immunoprecipitated by anti-FLAG Ab and subsequently analyzed by Western blots using anti-GPx8 and anti-FLAG Ab. GPx8 and its interacting proteins were precipitated by anti-GPx8 Ab and analyzed by indicated Abs.

5-2 LDH is a marker of cell lysis, not specifically of pyroptosis. The presence of pyroptosis should converge towards a classical activation rather than non classical.

Response: We have corrected the statement in the manuscript that LDH is a cell lysis marker instead of a pyroptosis marker. However, the cell lysis induced by LPS transfection is indeed a pyroptosis phenotype, as showed by many publications (Kayagaki, Wong et al., 2013, Shi et al., 2014). This is also a caspase-4/11 specific phenotype, which does not require caspase-1 to induce pyroptosis. On the contrary, IL-1 β production still requires caspase-1 (Man et al., 2017).

6-1 Fig. 4 It is very easy to have a false positive result in co-immunoprecipitation. In the blots of 4A, the co-ip seem to be very weak and this is confirmed in the colocalization experiment in 4B. Here a quantification of the colocalization should be shown. Further it is not clear whether the cells that are shown have been activated with LPS (and pretreatment) and if so what is the level of colocalization of untreated cells.

Response: Thank you for the suggestions. We has quantified the co-localization (R. Fig. 12D) and improved the quality of immunofluorescence assays (R. Fig. 12). Original 4B are cells without treatment. We further showed the co-localization of untreated cells (R. Fig 12A), cells with LPS priming (R. Fig 12B), and cells with transfection of LPS (R. Fig 12C). It should be noticed that after caspase-4 activation, cells were undergoing pyroptosis leading to the decrease of co-localization as shown in R. Fig. 10C.

R. Fig. 12 Co-localization of endogenous caspase-4 and Myc tagged GPx8 (GPx8-Myc) by immunofluorescence assays. Co-localization was detected in THP-1 cells in the absence (A) or presence of LPS priming signal (B). (C) Co-localization was observed in THP-1 cells primed with LPS for 6 h and transfected with LPS for 4 h. (D) Co-localization was quantified by Pearson's correlation coefficient, which is used in statistics to measure how strong a co-localization is between two proteins. The formulas return a value between -1 and 1, where 1 indicates a strong positive relationship.

6-2 Why in figure 4C only one treatment with LPS was performed and not the pretreatment, or was it a transfection?

Response: In Fig 4C, co-localization is detected in untreated cells and cells primed with LPS. We have included the experimental details for Fig. 4C (R. Fig. 13).

R. Fig. 13 Interaction between GPx8 and caspase-4 in THP-1 cells in the presence or absence of priming LPS was demonstrated by *in situ* proximity ligation assay (PLA) (red dots, labeled by arrowheads). GPx8-Myc stably expressing THP-1 were used to detect the interaction between GPx8-Myc and endogenous caspase-4. Untreated cells (Ctrl) or cells primed with LPS (primed LPS) were incubated with anti-

Myc and anti-caspase-4 Abs according to the manufacturer's instructions and stained nucleus with DAPI (blue). Results were quantified by counting at least 5 different fields with an average of 300 cells. Reactions without primary Abs were used as the control for PLA assay (PLA Ctrl).

7 I have the impression that GPX8 major target is the classical activation of the inflammasome. Why haven't the authors analyzed also the binding to caspase 1? The finding that the mutants abolish the inhibitory activity does not imply that this occurs through caspase4 interaction.

Response: We have cleared the doubts by showing that GPx8 does not interact with caspase-1 (R. Fig. 11) and GPx8 was not directly involved in classical inflammasome activation (R. Fig. 10).

8 Fig. 6. The experiments with the casp1/4 inhibitor VX-765 are very interesting but do not rule out the possibility that GPX8 actually does not interact through caspase 4.

Response: Thank you for the positive comments. After we have excluded the possibility that caspase-1 may be involved, this experiment demonstrated the rescuing phenotypes of over-activation of noncanonical inflammasome under Gpx8 deficiency. VX-765 has been used in clinical trials for other inflammatory diseases but not colitis. We would like to show that it may also be a potential drug for colitis patients by targeting noncanonical inflammasome pathway.

9 Fig 7A the control of uninflamed area is missing.

Response: We also analyzed the non-inflamed area from healthy individuals. Similar to non-inflamed regions in UC patients, GPx8 expressed in both CD86 and CD163 positive cells (R. Fig. 14).

R. Fig. 14 Colon biopsy of non-inflamed sections from health individuals were stained with markers for pan-macrophage, CD68, or monocytes/macrophages activating marker, CD163, combined with anti-GPx8. Quantification of double positive cells were calculated from three different individuals and presented as the mean \pm SEM of the percentage of double positive cells. The total cell numbers are the sum of double and single positive only cells.

10 What is the expression of caspase 1 and activated gasdermin D? While the downregulation of GPx8 is clear, the upregulation of caspase 4 is not, further why haven't the authors analyzed the activated form of caspase 4?

Response: After excluding the possibility of caspase-1 involvement, we focused on caspase-4 and its downstream proteins. In non-inflamed colon tissues, the pyroptosis and caspase-4 activation should not be detected because they are still normal tissues. As shown in R. Fig. 15, the cleavage form of GSDMD was not detected in samples tested. This result is not surprised because they are normal part of colon in UC patients. Since the cleavage of GSDMD was not detected in non-inflamed regions, there should be no pyroptosis and activated form of caspase-4. However, we observed higher levels of caspase-4 expression in those tissues (Fig. 7), which is consistent with the previous report demonstrating upregulation of caspase-4 expression in IBD patients (Flood, Oficjalska et al., 2015). Moreover, the most important finding is that GPx8 expression in normal colon tissue of these patients is reduced, implicating the significance of GPx8 in UC. The accumulation of cell death and immune cell infiltration in inflamed colon tissue prevents any comparison of the amount of GPx8 expression in inflamed tissue from UC patients and tissue from healthy individuals, who do not have inflamed parts. It is also difficult to quantify the amount of caspase-4 in inflamed tissue. We therefore only quantified levels of caspase-4 and GPx8 in non-inflamed regions.

R. Fig. 15 Protein levels of full-length and cleavage GSDMD in non-inflamed biopsy specimens from healthy controls (Ctrl) and UC patients (UC) were demonstrated by immunoblots. THP-1 cells were activated by LPS priming and then transfected with LPS (TF LPS) as a positive control demonstrating the cleavage form of GSDMD. PMA, PMA differentiated THP-1 cells; LPS, THP-1 cells primed with LPS; TF LPS, LPS transfected THP-1 cells.

References

- Bader JE, Enos RT, Velazquez KT, Carson MS, Nagarkatti M, Nagarkatti PS, Chatzistamou I, Davis JM, Carson JA, Robinson CM, Murphy EA (2018) Macrophage depletion using clodronate liposomes decreases tumorigenesis and alters gut microbiota in the AOM/DSS mouse model of colon cancer. *Am J Physiol Gastrointest Liver Physiol* 314: G22-G31
- Demon D, Kuchmiy A, Fossoul A, Zhu Q, Kanneganti TD, Lamkanfi M (2014) Caspase-11 is expressed in the colonic mucosa and protects against dextran sodium sulfate-induced colitis. *Mucosal Immunol*
- Flood B, Oficjalska K, Laukens D, Fay J, O'Grady A, Caiazza F, Heetun Z, Mills KH, Sheahan K, Ryan EJ, Doherty GA, Kay E, Creagh EM (2015) Altered expression of caspases-4 and -5 during inflammatory bowel disease and colorectal cancer: Diagnostic and therapeutic potential. *Clin Exp Immunol* 181: 39-50
- Kayagaki N, Wong MT, Stowe IB, Ramani SR, Gonzalez LC, Akashi-Takamura S, Miyake K, Zhang J, Lee WP, Muszynski A, Forsberg LS, Carlson RW, Dixit VM (2013) Noncanonical inflammasome activation by intracellular LPS independent of TLR4. *Science* 341: 1246-9
- Man SM, Karki R, Briard B, Burton A, Gingras S, Pelletier S, Kanneganti TD (2017) Differential roles of caspase-1 and caspase-11 in infection and inflammation. *Sci Rep* 7: 45126
- Shi J, Zhao Y, Wang Y, Gao W, Ding J, Li P, Hu L, Shao F (2014) Inflammatory caspases are innate immune receptors for intracellular LPS. *Nature* 514: 187-92
- Weisser SB, van Rooijen N, Sly LM (2012) Depletion and reconstitution of macrophages in mice. *J Vis Exp*: 4105

2nd Editorial Decision

22 July 2019

Thank you for the submission of your revised manuscript to EMBO Molecular Medicine. We have now heard back from the two referees whom we asked to evaluate your manuscript a second time. You will see that while ref. #1 is now satisfied, ref. #2 remains with questions. We would like to give you a last chance to address these issues to the satisfaction of the referee. Please bear in mind that revising now will not ascertain publication later on and your revision will be reviewed once more from the same set of referee(s).

I look forward to seeing a revised form of your manuscript within 3 months. As before, please note that EMBO Molecular Medicine strongly supports a single round of revision and that, as acceptance or rejection of the manuscript will depend on another round of review, your responses should be as complete as possible.

***** Reviewer's comments *****

Referee #1 (Remarks for Author):

The authors repeated some of the experiments and adequately addressed most of my concerns. As such, I have no further comments.

Referee #2 (Remarks for Author):

This is a very interesting paper associating the non canonical activation of the inflammasome to GPX8 and caspase 4/11. Altogether the data are not novel for each individual player but have never been linked together. I have some remarks.

There are many English typos and grammatical errors that should be corrected.

The difference when using macrophages from WT or GPX8^{-/-} mice is really minute as compared to the whole KO suggesting that there are other cells contributing to this effect or that rather than macrophages maybe inflammatory monocytes (or monocytes in general) should be transferred. Maybe BMDM do not really resemble cells that are conditioned by the local inflammatory environment after monocyte differentiation in situ. The authors should evaluate this possibility.

In addition, in the gut the distinction between macrophages and DCs is very subtle and CX3CR1⁺ cells share similarities between macrophages and DCs. The authors should analyze the expression of GPX8 in the different mononuclear phagocytes present in the gut (based on CX3CR1, CD11c, CD11b and CD103 markers).

Not clear to me why TLR4 should not be required for the effect. How do the authors envisage LPS is internalized? They performed the experiments with transfected LPS. Why do they need to transfect LPS? Fig. 3 would need a schematic to explain how the experiments were performed. It looks to me as if the expression of human GPX8 in GPX8^{-/-} BMDM induces also a change in the expression of procaspase 11. Can the authors quantify the blots, please? The same seems to occur in vivo.

The authors should also evaluate whether they can observe similar caspase activation in GPX8^{-/-} mice by doing the opposite (LPS first and then poli:IC or Poli:IC and poli:IC to evaluate whether this characteristic is exclusive to LPS).

The sentence: ' GPx8 deficiency promotes noncanonical but not canonical NLRP3 inflammasome activation.' is not correct as GPX8 deficiency does not promote, rather it does not contrast NLRP3 inflammasome.

It is not clear to me why there is no improvement in DSS colitis in WT mice treated with VX-765. Does this mean that the GPX8 and Caspase 4/11 does not normally play a role in colitis? Is the effect of GPX8 something that would not be evidenced and is just an epiphenomenon of the lack of GPX8? Or is the effect only transient in the first days of colitis (as the clinical score is modified only early? This should be clarified.

It looks to me that in UC patients there is a clear reduction of GPX8 but not necessarily and increase in Caspase 4. Maybe GPX8 acts also through another mechanism. Could the authors pair the expression of GPX4 with that of Caspase 4 for each individual patient (and control) to evaluate the relationship?

Is there any polymorphism in patients in genes for either Caspase 4 or GPX8 at the level of the cysteines found to be involved in the covalent linking?

2nd Revision - authors' response

13 September 2019

Referee #1 (Remarks for Author):

The authors repeated some of the experiments and adequately addressed most of my concerns. As

such, I have no further comments.

Referee #2 (Remarks for Author):

This is a very interesting paper associating the non canonical activation of the inflammasome to GPX8 and caspase 4/11. Altogether the data are not novel for each individual player but have never been linked together. I have some remarks.

Response: Thank you for considering that the paper is interesting. *GPx8*^{-/-} mice have never been characterized and we are the first one who identified the biological role of GPX8. We have faith in these novel findings. We shall clear up all the concerns raised.

1 There are many English typos and grammatical errors that should be corrected.

Response: Thank you for the suggestion. We have asked a native-English speaker to help edit the paper.

2 The difference when using macrophages from WT or GPX8^{-/-} mice is really minute as compared to the whole KO suggesting that there are other cells contributing to this effect or that rather than macrophages maybe inflammatory monocytes (or monocytes in general) should be transferred. Maybe BMDM do not really resemble cells that are conditioned by the local inflammatory environment after monocyte differentiation in situ. The authors should evaluate this possibility.

Response: In fact, the difference when using macrophages from WT or GPX8^{-/-} mice is similar as compared to the whole KO at the same period of time, 7 days. The difference should be much magnified if the experiment could be prolonged to 14 days. However, the experiment had to be stopped due to the limitation of the inhibition of macrophage generation (Fig 1c compared with Fig 2c). Moreover, monocytes recruited by chemokines differentiate locally into intestinal macrophages through stimulation of M-CSF. Thus, M-CSF-induced BMDMs, which we used in this study, are widely employed as an in vitro model for intestinal macrophages and for adoptive transfer for in vivo model [1-3]. Because it has been demonstrated that the adoptive transferred BMDMs are able to be recruited into tissue and differentiated into M1 and M2 macrophages [1], it is reasonable to use BMDMs for our adoptive transfer experiments. To exclude the possibility that WT and GPX8^{-/-} monocytes may be impaired in differentiating into different types of macrophages, we characterize the percentage of M1 and M2 macrophages differentiated from progenitors as previously described [4]. We performed regular BMDMs culture with M-CSF and these cells were classically activated (M1 condition) with LPS (100 ng/ml) + IFN- γ (20ng/mL) on day 7, or alternatively activated (M2 condition) with IL-4 (20ng/mL) or received media alone (M0 condition)[4]. After analysis of the M2 surface marker, CD206, or RNA expression of M1 markers, TNF- α and iNOS, we can distinguish M1 and M2 macrophages with different amount of M1 and M2 genes expression (R. Fig. 1A, C and D). No difference of CD206, TNF- α , and iNOS, expression in macrophages differentiated from either WT or GPX8^{-/-} mice (R. Fig. 1B, 1C and 1D), suggested WT or GPX8^{-/-} BMDMs have similar differentiation ability.

R. Fig. 1 Bone marrow derived progenitors isolated from WT or GPx8^{-/-} mice were in the culture condition with M-CSF and on day 7 cells were then classically activated (M1 condition, M1) with LPS (100 ng/ml) + IFN- γ (20ng/mL), alternatively activated (M2 condition, M2) with IL-4 (20ng/mL) or received media alone (M0 condition, M0) [4]. (A-B) After polarization, macrophages were stained with Abs (CD45, F4/80, CD11b and CD206) and analyzed by flow cytometry. Cells were gated as the CD45⁺F4/80⁺CD11b^{hi} population, and further analyzed with CD206 expression. (C-D) RNA expression of M1 markers, TNF- α (C) and iNOS (D), were showed by quantitative reverse transcriptase PCR.

3 In addition, in the gut the distinction between macrophages and DCs is very subtle and CX3CR1+ cells share similarities between macrophages and DCs. The authors should analyze the expression of GPX8 in the different mononuclear phagocytes present in the gut (based on CX3CR1, CD11c, CD11b and CD103 markers).

Response: In order to identify the cell population expressing GPx8, we analyzed GPx8 in various immune cells including splenocytes, T cells, B cells, BMDMs and bone marrow derived dendritic cells (BMDMs). Among all cell types tested, we detected GPx8 expressed only in macrophages (R. Fig. 2A). GPx8 expression is relatively low in DCs from mice or from two human individuals. We also confirmed CD markers of macrophages (F4/80^{high}/CD11b⁺/CD11c^{low}) and DCs (F4/80^{low}/CD11b⁺/CD11c^{high})(R. Fig. 2B), suggested that GPx8 expressed primarily in macrophages.

To further confirm the expression of GPx8 in gut, we also analyzed intestinal macrophages and Dc surface markers. Intestinal macrophages and DCs express an overlapping pattern of surface molecules. Intestinal DCs are defined by the expression of CD11c, CD103, major histocompatibility complex (MHC) class II and differential expression of CD11b [5]. In contrast to DCs, monocytes and macrophages in the intestine lack CD103 expression and are defined by expression of F4/80 and CX3CR1 [5]. Because intestinal CD103⁺, but not CX3CR1⁺, antigen sampling cells migrate in lymph and serve classical dendritic cell functions [6], we therefore analyzed GPx8 expression associated with CD103. By using IHC with Abs against CD103, F4/80 and GPx8, we found that GPx8 expression highly co-localized with F4/80 positive cells but much less correlated with CD103 positive cells (R. Fig. 2C, D and E), suggested that GPx8 predominantly expressed in macrophages but not DC populations in gut.

R. Fig. 2 GPx8 is expressed predominantly in macrophages. (A) GPx8 is expressed predominantly in macrophages and not in T, B, or dendritic cells. Cell-type specific expression of GPx8 was elucidated by immunoblotting. The black arrowhead denotes GPx8. The white arrowhead denotes non-specific bands. (B) BMDMs and Dcs isolated from WT or GPx8^{-/-} mice were stained with Abs (F4/80, CD11b and CD11c) and analyzed by flow cytometry. (C-D) Colon tissues were stained with Abs (GPx8, CD103 and F4/80) and analyzed by confocal microscopy. Quantification of double-positive cells of GPx8 and CD103 (C) or GPx8 and F4/80 (D) was calculated from 7 fields and is presented as the mean \pm SEM of the percentage of double-positive cells (E).

4 Not clear to me why TLR4 should not be required for the effect.

Response: It has been shown that noncanonical inflammasome activation by intracellular LPS is independent of TLR4 [7]. Since the TLR4 or other TLRs are only required for the priming signals for inducing pro-caspase-4/11 expression, we can easily replace the LPS priming signal with other TLR agonists, such as poly (I:C), a TLR3 agonist (Fig. 3C). To exclude the priming signals involved, no differential response of WT and GPx8^{-/-} BMDMs in cytokine production activated by different ligands of TLRs was observed (Sup. Fig. 2C and R. Fig. 3). We also revealed that the different priming signals displayed the same results in *in vivo* sepsis model (Fig. 3I and J), suggesting that the priming signal, such as TLR4, is not required for the effect of GPx8 deficiency.

R. Fig. 3 GPx8 is expressed predominantly in macrophages and does not interfere with the TLR signaling pathways. GPx8-deficiency has no effect on TLR signaling pathways. BMDMs from WT and GPx8^{-/-} mice were primed with IFN γ (250 U/ml) and then triggered by each ligand for TLRs for 16 h. Secreted cytokines TNF- α (A) and IL-6 (B) were quantified by ELISA as indicated.

5 How do the authors envisage LPS is internalized? They performed the experiments with transfected LPS. Why do they need to transfected LPS?

Response: Given that caspase-4/11 is the intracellular receptor of LPS [7, 8], we therefore followed the protocol published in Science 2013 (Kayagaki *et al.*) to perform the LPS transfection [7]. Induction of priming signals, such as TLR4 or TLR3, which are membrane receptors, only triggered IL-6 and TNF- α production but not caspase-4/11 activation. Both pyroptosis and IL-1 β production are dependent on caspase-4/11 activation by intracellular LPS. Therefore, the phenomena including the cleavage of caspase-4, pyroptosis and IL-1 β production all suggested that transfected LPS was internalized and activated caspase-4 (fig. 3A, B, and C).

6 Fig. 3 would need a schematic to explain how the experiments were performed.

Response: We included a flow chart of experimental procedures (R. Fig. 4).

R. Fig. 4 A flow chart of experimental procedures.

7 It looks to me as if the expression of human GPX8 in GPX8^{-/-} BMDM induces also a change in the expression of procaspase 11. Can the authors quantify the blots, please? The same seems to occur *in vivo*.

Response: Thank you for the suggestions. We also analyzed the expression of pro-caspase-11 in BMDMs with or without ectopically expressing human GPx8. No differential expression of pro-caspase-11 was observed (R. Fig. 5). After quantified the expression of caspase-11 in immunoblots, we concluded that there was also no difference of pro-caspase-11 expression between WT and GPx8^{-/-} BMDMs before LPS transfection (R. Fig. 6, lane 1 and 6). After LPS transfection, more IL-1b production of GPx8^{-/-} BMDMs compared with WT BMDMs. Since IL-1β known to be a potent inducer of IL-6 and TNF-α [9, 10], it leads to more IL-6 and TNF-α and downstream signaling, which primes more pro-caspase-11 expression (R. Fig. 6, lane 7 and 8). Similar pattern was also observed in WT BMDMs but displayed as a smaller wave (R. Fig. 6, lane 2 and 3). In *In vivo* setting, because the colon tissues were collected after DSS treatment, it is logical to expect that the activation of caspase-4 and more infiltrated immune cells may lead to more pro-caspase-11 and pro-caspase-1 production and activation in GPx8^{-/-} mice (Fig. 3F). We have incorporated R. Fig. 6 into the Revised Fig. 3F.

R. Fig. 5 RNA expression of *caspase-11* was not altered in BMDMs with or without ectopically expressing human GPx8 (hGPx8). WT or GPx8^{-/-} BMDMs were transduced with empty vector (EV) or hGPx8 expressing lentiviruses. Expression of caspase-11 mRNA in BMDMs were confirmed by quantitative reverse transcriptase PCR.

R. Fig. 6 GPx8 enhanced caspase-11 cleavage and activation. BMDMs from WT or GPx8^{-/-} mice were primed with LPS (lane 1 and 6) and then transfected with various amount of LPS (lane 2-5 and 7-10). Levels of processed caspase-1 and -11 in the supernatants (Sup) and full-length caspase-1, -11, and GAPDH in the cell lysates (Cell ext) were determined by immunoblotting. Fold change of protein expression was normalized with the internal control and indicated below each blot.

7 The authors should also evaluate whether they can observe similar caspase activation in GPX8^{-/-} mice by doing the opposite (LPS first and then poli:IC or Poli:IC and poli:IC to evaluate whether this characteristic is exclusive to LPS).

Response: As we mentioned previously, we used a nonlethal dose of poly(I:C) (LMW, 1 mg/kg) as primed signal and then re-challenged 6 h later with LPS (10 mg/kg) for sepsis model. Mortality doesn't occur in reverse order [11]. In addition, it has been clearly demonstrated that poly(I:C) does not induce caspase-4/11 activation and caspase-4/11 dependent sepsis phenotype *in vivo* [7].

8 The sentence: 'GPx8 deficiency promotes noncanonical but not canonical NLRP3 inflammasome activation.' is not correct as GPX8 deficiency does not promote, rather it does not contrast NLRP3 inflammasome.

Response: Thank you for the suggestion. We corrected the sentence.

9 It is not clear to me why there is no improvement in DSS colitis in WT mice treated with VX-765. Does this mean that the GPX8 and Caspase 4/11 does not normally play a role in colitis? Is the effect of GPX8 something that would not be evidenced and is just an epiphenomenon of the lack of GPX8? Or is the effect only transient in the first days of colitis (as the clinical score is modified only early)? This should be clarified.

Response: We clearly observed the difference by treating WT mice with VX-765. The treating group showed lower clinical scores on day 3 (P=0.006) and day 5 (P=0.0004) (R. Fig 7A). Despite the differences of clinical scores became less significant in the recovery stage, the mice produced lower IL-1b in colon of mice with VX-765 when they were sacrificed at the end of the experiment, suggested that inflammasome was activated at this stage and was inhibited by VX-765 (P=0.0005) (R. Fig. 7B). Although whether canonical inflammasome is pathogenic or beneficial in different IBD status remains uncertain, a consensus that over-activation of NLRP3 inflammasome has a major role in the pathogenesis of IBD and several autoinflammatory diseases has been reached in the field [12, 13]. Therefore, GPx8^{-/-} mice with over-activation of noncanonical inflammasome can be rescued by VX-765, indicated that Gpx8 is involved in this pathway. Further studies for the role of caspase-4/11 in IBD will be investigated in future.

R. Fig. 7 Caspase-4/11 inhibitor, VX-765, suppresses DSS-induced colitis in GPx8^{-/-} mice. Mice were injected intraperitoneally with VX-765 (20 mg/kg) under colitis model. (A) The clinical scores of mice induced with 4% DSS for 6 days (n = 10-11 per group). (B) Colon samples from mice treated with 4% DSS for 5 days (n = 6-10 per group). Productions of IL-1β and IL-6 in colon tissue lysates were demonstrated on day 14 after treatment. Data are presented as the mean ± SEM. NS: not significant (P > 0.05). *P < 0.05; **P < 0.01; ***P < 0.001 (unpaired Student's t-test).

10 It looks to me that in UC patients there is a clear reduction of GPX8 but not necessarily and increase in Caspase 4. Maybe GPX8 acts also through another mechanism. Could the authors pair the expression of GPX4 with that of Caspase 4 for each individual patient (and control) to evaluate the relationship?

Response: Thank you for the suggestions. After paired the expression of GPx8 and caspase-4 in each individuals, we can clearly detect the correlation in patients, where a low Gpx8 expression links to a high caspase-4 expression (P=0.0043), but not in normal controls (R Fig. 8). It suggested that less GPx8 expression correlated with more caspase-4 expression in UC patients.

R. Fig. 8 Lower GPx8 and higher caspase-4 expression in colon tissue of ulcerative colitis patients. Relative expression levels of GPx8 and caspase-4 (Casp4) in non-inflamed biopsy specimens from healthy controls (Ctrl) (n = 14) and UC patients (n = 25) are demonstrated by immunoblots. Data are normalized to the mean of healthy controls and presented as the mean ± SEM. GPx8 and caspase-4 expression paired in each individual are linked by lines. P value was analyzed by Fisher's exact test.

11 Is there any polymorphism in patients in genes for either Caspase 4 or GPX8 at the level of the cysteines found to be involved in the covalent linking?

Response: Thank you for the suggestions. Genome-wide association studies (from GWAS Central) showed there is no SNP associated with any cysteine in both genes.

- Jung, S.B., et al., *Reduced oxidative capacity in macrophages results in systemic insulin resistance*. Nat Commun, 2018. 9(1): p. 1551.
- Bain, C.C. and A.M. Mowat, *Macrophages in intestinal homeostasis and inflammation*. Immunol Rev, 2014. 260(1): p. 102-17.

3. Weisser, S.B., N. van Rooijen, and L.M. Sly, *Depletion and reconstitution of macrophages in mice*. J Vis Exp, 2012(66): p. 4105.
4. Jablonski, K.A., et al., *Novel Markers to Delineate Murine M1 and M2 Macrophages*. PLoS One, 2015. **10**(12): p. e0145342.
5. Gross, M., T.M. Salame, and S. Jung, *Guardians of the Gut - Murine Intestinal Macrophages and Dendritic Cells*. Front Immunol, 2015. **6**: p. 254.
6. Schulz, O., et al., *Intestinal CD103+, but not CX3CR1+, antigen sampling cells migrate in lymph and serve classical dendritic cell functions*. J Exp Med, 2009. **206**(13): p. 3101-14.
7. Kayagaki, N., et al., *Noncanonical inflammasome activation by intracellular LPS independent of TLR4*. Science, 2013. **341**(6151): p. 1246-9.
8. Shi, J., et al., *Inflammatory caspases are innate immune receptors for intracellular LPS*. Nature, 2014. **514**(7521): p. 187-92.
9. Tosato, G. and K.D. Jones, *Interleukin-1 induces interleukin-6 production in peripheral blood monocytes*. Blood, 1990. **75**(6): p. 1305-10.
10. Huang, J., et al., *Recruitment of IRAK to the interleukin 1 receptor complex requires interleukin 1 receptor accessory protein*. Proc Natl Acad Sci U S A, 1997. **94**(24): p. 12829-32.
11. Wang, A., et al., *Specific sequences of infectious challenge lead to secondary hemophagocytic lymphohistiocytosis-like disease in mice*. Proc Natl Acad Sci U S A, 2019. **116**(6): p. 2200-2209.
12. Reinecker, H.C., et al., *Enhanced secretion of tumour necrosis factor-alpha, IL-6, and IL-1 beta by isolated lamina propria mononuclear cells from patients with ulcerative colitis and Crohn's disease*. Clin Exp Immunol, 1993. **94**(1): p. 174-81.
13. Liu, L., et al., *The Pathogenic Role of NLRP3 Inflammasome Activation in Inflammatory Bowel Diseases of Both Mice and Humans*. J Crohns Colitis, 2017. **11**(6): p. 737-750.

3rd Editorial Decision

17 October 2019

Thank you for the submission of your revised manuscript to EMBO Molecular Medicine. We have now received the enclosed report from the referee who that was asked to re-assess it. As you will see the reviewer is now globally supportive and I am pleased to inform you that we will be able to accept your manuscript pending minor editorial amendments and addressing the comments from referee 2.

Please submit your revised manuscript within two weeks. I look forward to seeing a revised form of your manuscript as soon as possible.

***** Reviewer's comments *****

Referee #2 (Remarks for Author):

Although I am satisfied with the revision, the authors should include at least figure 2, figure 4 and figure 8 for the reviewer into the manuscript. These have been asked for to support the data or to make them clearer to the reader and not only to the reviewer.

3rd Revision - authors' response

24 October 2019

We appreciate that both reviewers agree with us to publish this paper. We have incorporated the comments from reviewer 2 into the main text. We have also fully addressed your requests.

Corresponding Author Name: Wen-Hua Lee and Jye-Lin Hsu

Manuscript Number: EMM-2018-09386-V3